# CD200–CD200R immune checkpoint engagement regulates ILC2 effector function and ameliorates lung inflammation in asthma

Pedram Shafiei-Jahani[1], Doumet Georges Helou [1], Benjamin P. Hurrell[1], Emily Howard[1], Christine Quach [1], Jacob D. Painter[1], Lauriane Galle-Treger[1], Meng Li[2], Yong-Hwee Eddie Loh[2] & Omid Akbari [1]✉

The prevalence of asthma and airway hyperreactivity (AHR) is increasing at an alarming rate. Group 2 innate lymphoid cells (ILC2s) are copious producers of type 2 cytokines, which leads to AHR and lung inflammation. Here, we show that mouse ILC2s express CD200 receptor (CD200R) and this expression is inducible. CD200R engagement inhibits activation, proliferation and type 2 cytokine production, indicating an immunoregulatory function for the CD200–CD200R axis on ILC2s. Furthermore, CD200R engagement inhibits both canonical and non-canonical NF-κB signaling pathways in activated ILC2s. Additionally, we demonstrate both preventative and therapeutic approaches utilizing CD200R engagement on ILC2s, which lead to improved airway resistance, dynamic compliance and eosinophilia. These results show CD200R is expressed on human ILC2s, and its engagement ameliorates AHR in humanized mouse models, emphasizing the translational applications for treatment of ILC2-related diseases such as allergic asthma.

[1] Department of Molecular Microbiology and Immunology, Keck School of Medicine, University of Southern California, Los Angeles, CA, USA. [2] Norris Medical Bioinformatics, Keck School of Medicine, University of Southern California, Los Angeles, CA, USA. ✉email: akbari@usc.edu

Allergic asthma is an inflammatory disorder of the airways that is characterized by bronchoconstriction, bronchial hyper-responsiveness, and tissue remodeling[1]. The increasing prevalence of this chronic disorder has fueled efforts to better understand the immunopathogenesis, discover novel biomarkers, and design effective strategies to directly modulate the activity driving type 2 immune cells[2]. The recent discovery of lineage negative group 2 innate lymphoid cells (ILC2s) has underscored the role of innate immunity in both initiation and perpetuation of asthma. ILC2s are rapid and proficient producers of type 2 cytokines such as IL-5 and IL-13[3]. These pluripotent type 2 cytokines play a central role in exacerbation of pulmonary inflammation. For example, IL-5 leads to eosinophilia by promoting recruitment, maturation, activation, and survival of bone marrow-derived eosinophils[4]. IL-13 causes both goblet cell hyperplasia and bronchiole smooth muscle contraction, which together lead to narrowing of the airways and difficulty breathing[5,6]. Pulmonary ILC2s are located near the basement membrane subjacent to the epithelium layer, less than 70 μm away from the bronchioles[7]. The strategic positioning of pulmonary ILC2s enables their rapid inflammatory response to alarmins such as IL-25, IL-33, and thymic stromal lymphopoietin (TSLP), and supports their role as the earliest inducers of type 2 inflammation in the lungs[8].

CD200 receptor (CD200R) is an immunoregulatory receptor prominently expressed in the lungs, and mainly reported among alveolar macrophages, neutrophils, mast cells[9–11]. Previous studies have shown CD200-deficient mice have increased pulmonary alveolar macrophage activity in a mouse influenza model, which leads to increased mortality and hindered resolution of pulmonary inflammation[9]. The corresponding ligand of CD200R, CD200, has no known signaling motif and is primary expressed by the pulmonary epithelial cells[12–14]. Near the pulmonary epithelial surface, modulation of immunological homeostasis is critical for maintenance of tolerance, tissue integrity, and proper lung function. Thus, the CD200–CD200R axis is considered an important immunological checkpoint with a pivotal role in maintenance of immune tolerance. However, the expression, role, and mechanism of CD200R signaling in ILC2s at this important interface remains to be described. Further investigation of the CD200–CD200R pathway will not only advance our understanding of asthma pathogenesis and tolerance, but also provide the rationale for novel targeted immunotherapeutic strategies.

In this study, we evaluate the mechanism, signaling, and therapeutic potentials of CD200R engagement on ILC2s in the context of allergic asthma and airway hyper-reactivity (AHR). We show that both peripheral blood human and lung-derived mouse ILC2s express CD200R, and this expression is further increased by IL-33. CD200R engagement reduces activation, decreases proliferation, and inhibits type 2 cytokine production in activated ILC2s. CD200R engagement inhibits both the canonical and noncanonical NF-κB pathways in activated pulmonary ILC2s, as evidenced by downregulation of pIKKα/β, *Nfkb1*, and *Rela* (p65), as well as *Nfkb2* (p52) and *Relb*. Utilizing CD200-Fc (CD200-Fc chimeric protein), we demonstrate the preventative and therapeutic role for CD200R engagement on ILC2s, resulting in reduced airway resistance, dampened eosinophilia, and improved lung dynamic compliance. The observed therapeutic effects of CD200R engagement are ILC2-dependent and validate the clinical relevance of our findings in *Alternaria alternata*-induced AHR. Importantly, we utilize preventative and therapeutic humanized mice models to highlight the efficacy of CD200R agonistic treatment in human ILC2-mediated AHR. Our results point to CD200R as an important regulator of ILC2s, thus providing insights into the role of CD200R in ILC2-driven

pulmonary inflammation and anti-CD200R as a promising treatment option for asthma and lung inflammation.

## Results

### CD200R is constitutively expressed and induced by IL-33 on mouse ILC2s.

Han et al. in a genome-wide analysis of 88,486 patients with asthma and 447,859 healthy donors using data from UK Biobank and the Trans-National Asthma Genetic Consortium[15], identified several human loci among asthmatic patients and bioinformatically showed variants of candidate causal genes, such as *CD200R1* that could potentially be associated with asthma immunopathogenesis. Since ILC2s play a significant role in initiating and perpetuating pulmonary inflammation, we began exploring whether CD200R is expressed on pulmonary ILC2s. A cohort of WT mice were challenged with the alarmin IL-33 (0.5 μg) or PBS intranasally (i.n.) on days 1, 2, and 3 (Fig. 1a). On the fourth day, ILC2s from the lungs were isolated and analyzed by flow cytometry and gated as lineage⁻ CD45⁺ CD127⁺ and ST2⁺ (Fig. 1b). Analysis of pulmonary ILC2s revealed that both naïve and IL-33-activated ILC2s (aILC2s) express CD200R at mRNA and protein levels (Fig. 1c, d). Although CD200R is expressed on naïve ILC2s, this expression is further inducible in vivo by IL-33 (Fig. 1d). The basal expression of CD200R in ILC2s and CD11b⁺F4/80⁺ macrophages (Supplementary Fig. 1a) in the lungs appear to be comparable (Supplementary Fig. 1b). Interestingly, CD200R is inducible by IL-33 in ILC2s, but not in macrophages. Moreover, blocking the ST2 receptor abrogates induction of CD200R by IL-33 in ILC2s (Supplementary Fig. 2). Single cell RNA (scRNA) analysis further revealed that ILC2s that highly express CD200R (dashed line) have abrogated total gene expression, as indicated by the smaller size of the cells (Fig. 1c) and correlation plots of CD200R versus total genes expression in naïve (Supplementary Fig. 3a) and IL-33-activated ILC2s (Supplementary Fig. 3b). Expression landscape of CD200R across all ILC2s (Supplementary Fig. 3c) further displays a high basal mRNA expression level, as well as a gradual expression increase among CD200R expressing ILC2s. In order to determine the kinetics of CD200R induction by IL-33 at a protein level, we next sorted naïve ILC2s from a cohort of WT mice and cultured them in presence of IL-2 and IL-7. The pulmonary ILC2s were subsequently ex vivo stimulated with IL-33, and CD200R expression was analyzed by flow cytometry over time at 0, 24, 48, and 72 h (Fig. 1e). We observed that CD200R expression was increased over time and reached statistical significance after 24 h of ex vivo IL-33 stimulation (Fig. 1f). To investigate the effect of CD200R engagement on ILC2s, we subsequently measured the cytokine secretion levels of ILC2s in the presence of CD200-Fc (10 μg/mL) or isotype control. Freshly isolated in vivo IL-33-activated pulmonary ILC2s were cultured ex vivo for 48 h with either CD200-Fc or corresponding isotype controls. The cytokine secretion levels were then measured in the cell culture supernatants by Luminex assay. Cytokine levels of IL-5, IL-6, IL-9, IL-13, and GM-CSF were diminished by CD200R engagement on activated ILC2s compared to controls (Fig. 1g). Moreover, secreted IL-10 levels were augmented after CD200R stimulation of ILC2s (Fig. 1g). Taken together, these results suggest that CD200R is expressed on lung-derived naïve and IL-33-activated ILC2s and acts as an inhibitory cell surface receptor.

### CD200R engagement inhibits ILC2 activation, proliferation, and function.

Since we observed reduced total gene expression in CD200R expressing ILC2s, we next explored the intracellular influences of CD200R on ILC2s. In line with the reduced total gene expression, scRNAseq analysis suggested reduced mRNA expression of *Il5* (Fig. 2a) and *Il13* (Fig. 2c). In order to determine the influence of CD200R engagement, we isolated in vivo

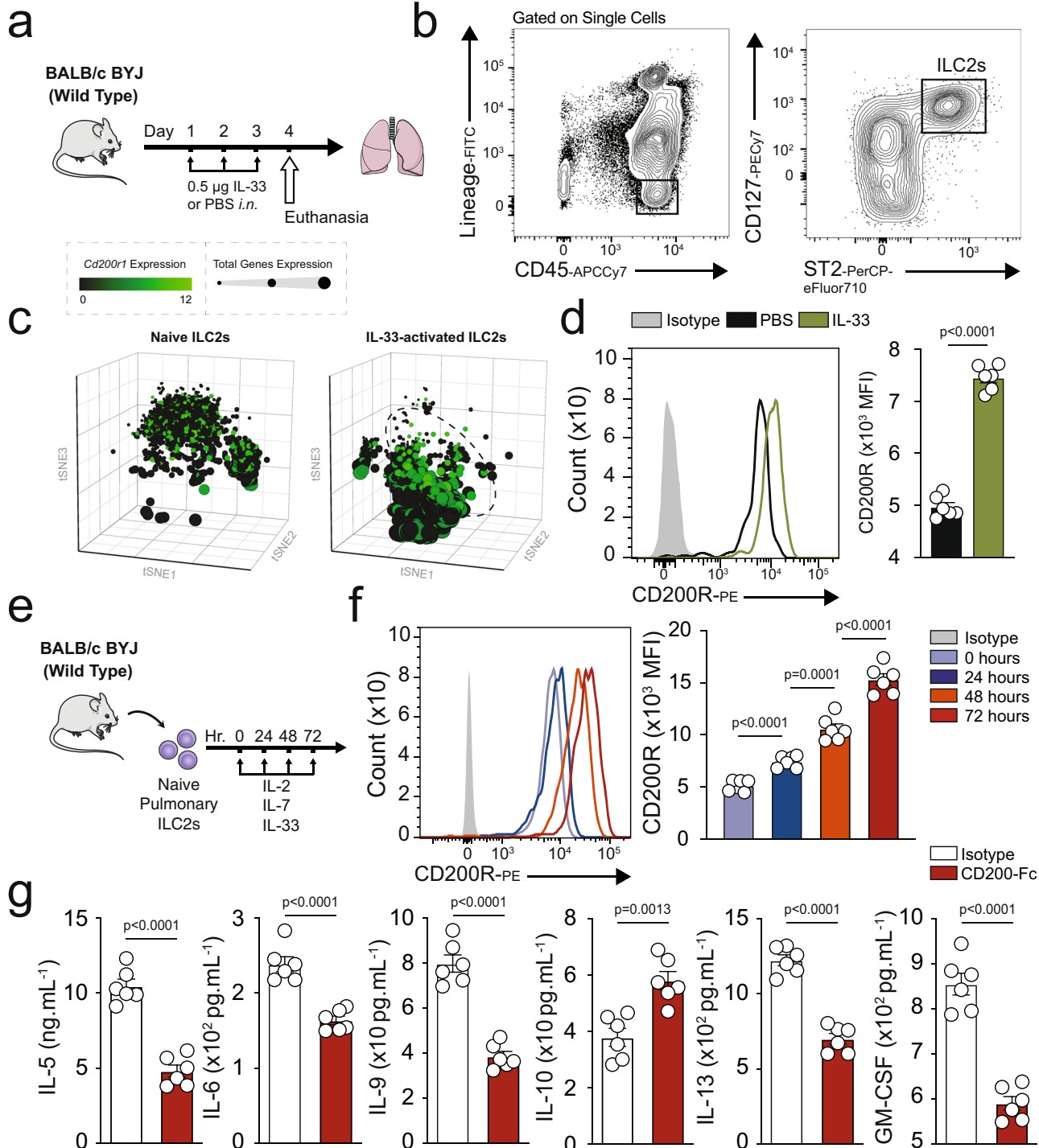

**Fig. 1 Mouse ILC2s express CD200R and this expression is inducible by IL-33. a** A cohort of WT mice were challenged with recombinant mouse (rm) IL-33 (0.5 μg) or PBS intranasally (i.n.) on days 1, 2, and 3. The mice were euthanized on day 4 and the lung was isolated, as shown in the timeline. **b** The gating strategy of ILC2s identified as Lin⁻CD45⁺CD127⁺ST2⁺ cells. **c**, **d** mRNA (tSNE plot of 4474 naïve and 3390 activated ILC2s; dot size is indicative of the total gene expression level in each cell; dashed line encloses cells with highest CD200R expression) and protein expression levels of CD200R in both naïve and IL-33-activated ILC2s in the lungs. Corresponding quantitation of CD200R expression shown as MFI $+/-$ SEM, $n = 6$ mice. **e** Naïve pulmonary ILC2s were sorted and subsequently cultured with rmIL-2 and rmIL-7 and rmIL-33 for 24, 48, and 72 h. **f** Freshly isolated ILC2s at 0 h and ex vivo activated ILC2s were analyzed by flow cytometry as indicated in the scheme and the kinetics of CD200R induction by rmIL-33 is shown. **g** Freshly sorted naïve and activated pulmonary ILC2s were cultured in the presence of rmIL-2 and rmIL-7 stimulated with CD200-Fc (10 μg/mL) or isotype control for 48 h. The levels of IL-5, IL-6, IL-9, IL-10, IL-13, and GM-CSF were measured by Luminex on the culture supernatants, $n = 6$ mice. Statistical analysis, two-tailed student's $t$-test or one-way ANOVA followed by Tukey post-hoc tests; Data are shown as means ± SEMs and are representative of three individual experiments. Mouse and lung images are sourced through an open access license from Servier Medical Art.

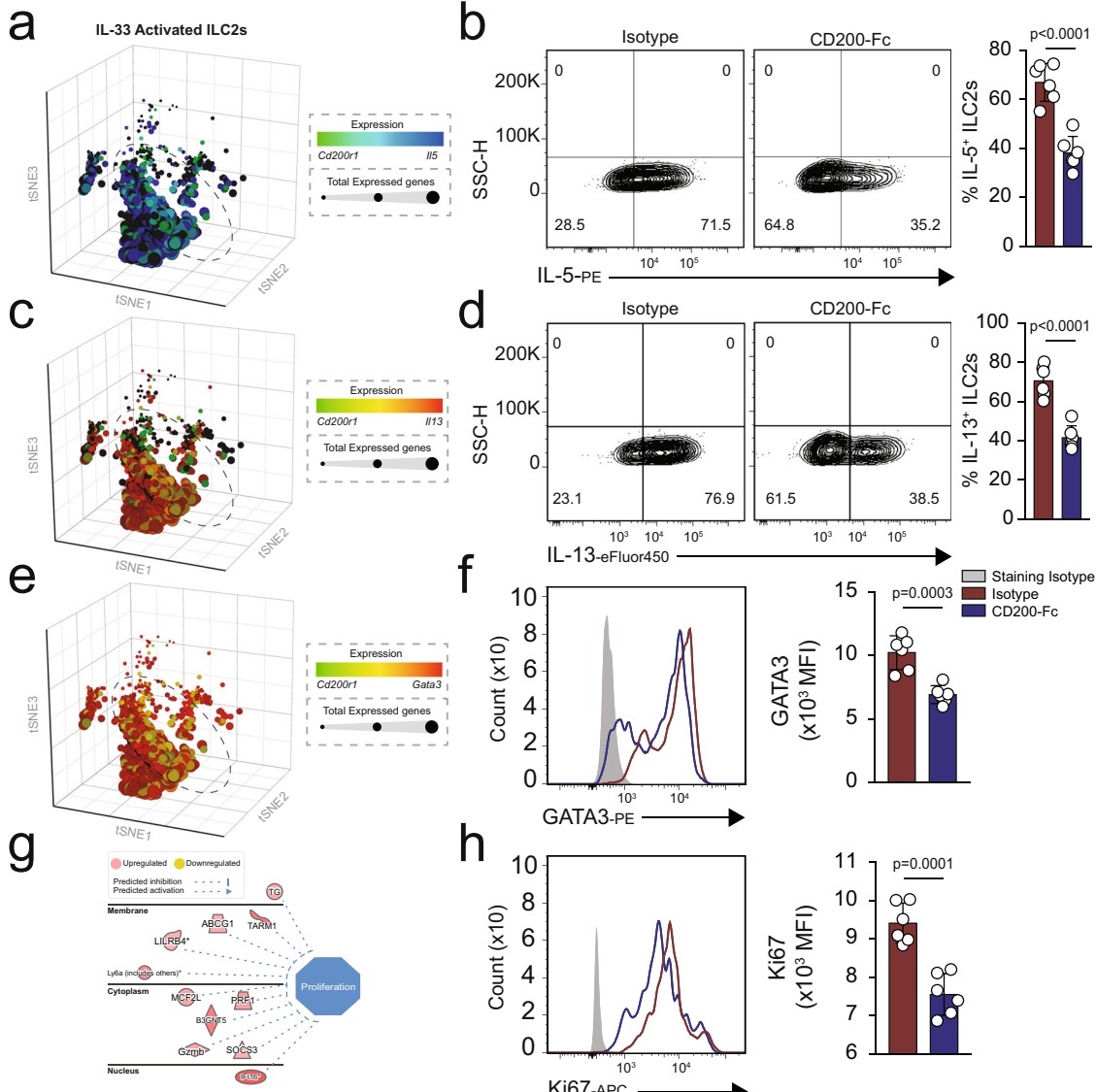

**Fig. 2 CD200R engagement inhibits ILC2 activation and effector function. a** A cohort of WT mice were challenged with recombinant mouse (rm) IL-33 (0.5 μg in 50 μL) for three consecutive days, and in vivo pulmonary ILC2s isolated on the fourth day, n = 6 mice. Activated ILC2s were subsequently cultured in the presence of recombinant mouse rmIL-2 and rmIL-7, and stimulated with CD200-Fc (10 μg/mL) or isotype control for 24 h. **a** scRNAseq profiles of 3390 IL-33-activated ILC2s from WT mice. Cells are colored based on expression levels of *Cd200r1* and *Il5*. Dot size is indicative of the total gene expression level of each cell. Dashed lines enclose cells with highest CD200R expression. **b** Representative flow cytometry plots of intracellular IL-5 protein expression levels in ILC2s cultured with CD200-Fc or isotype control. **c** t-Distributed stochastic neighbor embedding (tSNE) of 3390 ILC2s colored based on expression levels of *Il13* and *Cd200r1*. **d** intracellular protein expression levels of IL-13 in cultured ILC2s. **e** tSNE plots colored based on expression levels of C*d200r1* and *Gata3* in single ILC2s. **f** Intranuclear protein expression level of transcription factor GATA3 in cultured ILC2s. In vivo-activated ILC2s were cultured in presence of rmIL-2, rmIL-7, and rmIL-33 and stimulated with CD200-Fc (10 μg/mL) or isotype control for 24 h. Total RNA was isolated and sequenced. **g** Network analysis of upregulated (red) and downregulated (green) genes predicted to influence cell proliferation in cultured ILC2s. **h** Representative flow cytometry plots of intranuclear Ki67 levels in cultured ILC2s. Quantitation of protein expression levels are presented as MFI +/− SEM, n = 6 mice. Statistical analysis, two-tailed student's t-test (**b**, **d**, **f**, and **h**).

activated ILC2s and stimulated them ex vivo with CD200-Fc or isotype control. After 24 h, we measured the intracellular cytokine levels by flow cytometry. We observed that CD200R significantly abrogated both IL-5 (Fig. 2b) and IL-13 (Fig. 2d). ScRNAseq data also suggested reduced mRNA of the transcription factor *Gata3* (Fig. 2e). In confirmation with these results, intranuclear GATA3 protein levels were reduced by CD200R engagement (Fig. 2f). However, CD200R engagement did not polarize ILC2s toward either an ILC1 or ILC3 phenotype, as evident by lack of T-bet (Supplementary Fig. 4a) and RORγt (Supplementary Fig. 4b), respectively. In order to determine potential mechanisms of IL-10

induction, we next assessed expression levels of transcription factors FOXP3, Blimp-1, and c-Maf by flow cytometry. CD200R engagement was found to upregulate expression of Blimp-1, but not cMaf and FOXP3 (Supplementary Fig. 4c–e). Next, we ex vivo cultured activated pulmonary ILC2s from WT mice in presence of CD200-Fc or isotype control for 24 h, and subsequently performed RNA-sequencing. Network analysis of RNA-seq data revealed lower levels of proliferation in CD200-Fc-treated ILC2s compared to the isotype-treated cells (Fig. 2g). The flow cytometry data further confirmed reduced proliferation upon CD200R engagement, as indicated by reduced intranuclear Ki67

levels (Fig. 2h). Collectively, these results suggest CD200R is a potent inhibitory receptor of ILC2s capable of hindering activation, reducing proliferation, and modulating effector function.

**CD200R engagement has a protective effect against IL-33 induced AHR.** Next, we explored whether CD200R stimulation can prevent IL-33-induced airway hyper-reactivity and lung inflammation by comparing CD200-Fc with isotype treated control in WT mice. The mice were challenged with IL-33 (0.5 μg) or PBS intranasally (i.n.) and treated with CD200-Fc or isotype control i.n. on days 1, 2, and 3 (Fig. 3a). On the fourth day, lung function was measured by direct measurement of lung resistance and dynamic compliance in anesthetized tracheostomized mice, in which mice were mechanically ventilated and sequentially challenged with aerosolized increasing doses of methacholine. After measurements of AHR, the trachea was cannulated and the bronchoalveolar lavage (BAL) fluid was collected and analyzed by flow cytometry (Supplementary Fig. 5). Administration of IL-33 i.n. significantly increased lung resistance in isotype treated cohort but not in the CD200-Fc treated cohort, suggesting that CD200R engagement can prevent IL-33-induced AHR (Fig. 3b). In agreement with lung resistance findings, the results of dynamic compliance showed an improved response in CD200-Fc-treated cohort compared to isotype treated group (Fig. 3c). IL-33 treatment significantly increased the total numbers of $CD45^+$ leukocytes (Fig. 3d), $CD3^+$ T cells (Fig. 3e), and neutrophils (Fig. 3f) in the BAL fluid of isotype-treated control when compared to the CD200-Fc treated mice. Importantly, lung inflammation was impeded by CD200R stimulation, as indicated by reduced eosinophilia (Fig. 3g). Analysis of the lungs by flow cytometry revealed the number of pulmonary ILC2s was reduced in the CD200-Fc-treated cohort (Fig. 3h). In confirmation with these results, further histological analysis of the lungs (Fig. 3i) demonstrated that IL-33 challenge led to increased inflammatory cells (Fig. 3j), and thickening of the epithelium (Fig. 3k) in isotype control but not in the CD200-Fc treated cohort. Likewise, alcian blue/periodic acid–schiff (AB-PAS) staining revealed augmented mucous production in IL-33 challenged isotype-treated group compared to the IL-33 administered CD200-Fc-treated mice. These results indicate CD200-Fc treatment prevents IL-33-induced AHR and lung inflammation in WT mice.

In further investigation, we asked whether CD200R engagement can therapeutically reverse IL-33-induced AHR. A cohort of WT mice were then challenged with IL-33 or PBS intranasally (i.n.) for 3 days, and subsequently treated with CD200-Fc i.n. or isotype control on days 4–6 (Fig. 3l). Measurements of lung function and sample acquisition followed on day 7. CD200R engagement therapeutically improved lung function by decreasing lung resistance (Fig. 3m) and increasing dynamic compliance (Fig. 3n). Furthermore, CD200R stimulation effectively reduced pulmonary inflammation by reducing the total presence of $CD45^+$ leukocytes (Fig. 3o), $CD3^+$ T cells (Fig. 3p), neutrophils (Fig. 3q), as well as eosinophils (Fig. 3r) in the BAL fluid. Similarly, analysis of the lungs further revealed abrogated number of ILC2s in the CD200-Fc treated cohort (Fig. 3s). In concurrence with the reduction of AHR and eosinophilia, further histological analyses of the lungs revealed that CD200-Fc treatment after the intranasal challenge reduced the mucous production (Fig. 3t), abrogated number of infiltrating cells (Fig. 3u) and decreased airway epithelium thickness (Fig. 3v). Our results suggest that CD200R engagement in WT mice may translate to therapy for established allergic asthma and AHR.

**CD200R stimulation ameliorates ILC2-dependent AHR.** Following up with our results, we next questioned whether CD200-Fc treatment can prevent IL-33-induced AHR in absence of the adaptive immunity. We examined the effects of CD200R engagement in IL-33-induced AHR and inflammation using $Rag2^{-/-}$ mice that lack any mature B and T cells. A cohort of $Rag2^{-/-}$ mice received treatment with either CD200-Fc or isotype-matched control, and were challenged with IL-33 (0.5 μg) or PBS intranasally on days 1, 2, and 3. Measurements of lung function and sample acquisition followed on day 4 (Fig. 4a). Lung function analysis revealed that *i.n.* IL-33 administration increased lung resistance (Fig. 4b) and decreased dynamic compliance (Fig. 4c) in $Rag2^{-/-}$ mice that received the isotype control, but not in the mice that received the CD200-Fc treatment. IL-33 challenge led to BAL eosinophilia (Fig. 4d), increased BAL neutrophils (Fig. 4e), increased pulmonary ILC2s (Fig. 4f) and thus augmented lung inflammation in isotype-treated cohort, when compared to CD200-Fc-treated mice. In confirmation with these findings, histological results revealed increased goblet cell hyperplasia (Fig. 4g), augmented number of inflammatory cells (Fig. 4h), as well as increased thickening of the epithelium (Fig. 4i) in IL-33 challenged isotype treated group, but not in IL-33 challenged CD200-Fc-treated mice. These results indicate CD200R engagement prevents IL-33-driven AHR and lung inflammation in absence of adaptive immunity.

In order to determine whether CD200R engagement can therapeutically reserve ILC2-driven IL-33-induced AHR, we next intranasally challenged a cohort of $Rag2^{-/-}$ mice with IL-33 (0.5 μg) or PBS for three consecutive days. Subsequently, we intranasally treated the mice with either CD200-Fc or isotype control for three additional days. The lung function and sample were measured on day 7 (Fig. 4j). Consistent with the preventative results, CD200R engagement in absence of adaptive immunity continued to protect against airway resistance (Fig. 4k) and improved dynamic compliance (Fig. 4l). CD200R engagement also significantly ameliorated BAL eosinophilia (Fig. 4m), reduced presence of neutrophils in BAL (Fig. 4n), and abrogated the number of pulmonary ILC2 (Fig. 4o). Further histological analysis revealed that therapeutically treated $Rag2^{-/-}$ mice exhibited less mucous production (Fig. 4p), reduced presence of inflammatory cells (Fig. 4q) and reduced thickening of the epithelium (Fig. 4r), further confirming amelioration of lung inflammation via CD200R engagement. In continuation, we investigated whether CD200R engagement can prevent and reverse AHR and lung inflammation induced by a clinically relevant allergen—*Alternaria alternata*—that is known to trigger a type 2 innate immune response[16]. Consistent with our findings in the IL-33 models, anti-CD200 agonistic treatment both prevented and reversed *Alternaria alternata*-induced AHR and lung inflammation in $Rag2^{-/-}$ mice (Supplementary Fig. 6). Taken together, these results indicate CD200R-targeted therapy ameliorates ILC2-driven AHR and lung inflammation in absence of adaptive immunity.

**CD200R signaling inhibits canonical and non-canonical NF-κB pathways.** To further explore the molecular mechanisms associated with protective effects of CD200R-depedent inhibition of ILC2s, we isolated activated pulmonary ILC2s from WT mice and ex vivo stimulated them with CD200-Fc or isotype control (10 μg/mL) for 24 h. Subsequently, we performed RNA-sequencing (RNA-seq) analysis in order to quantify the transcriptomic landscape. Principal component analysis (Fig. 5a) and volcano plot (Fig. 5b) reveal CD200R engagement on ILC2s resulted in distinct transcriptomic states and differential modulation of 584 genes (385 downregulated, 194 upregulated, $p \leq 0.05$ calculated via gene set analysis (GSA) using lognormal with shrinkage, 1.5FC), further depicted as a heat map (Fig. 5c).

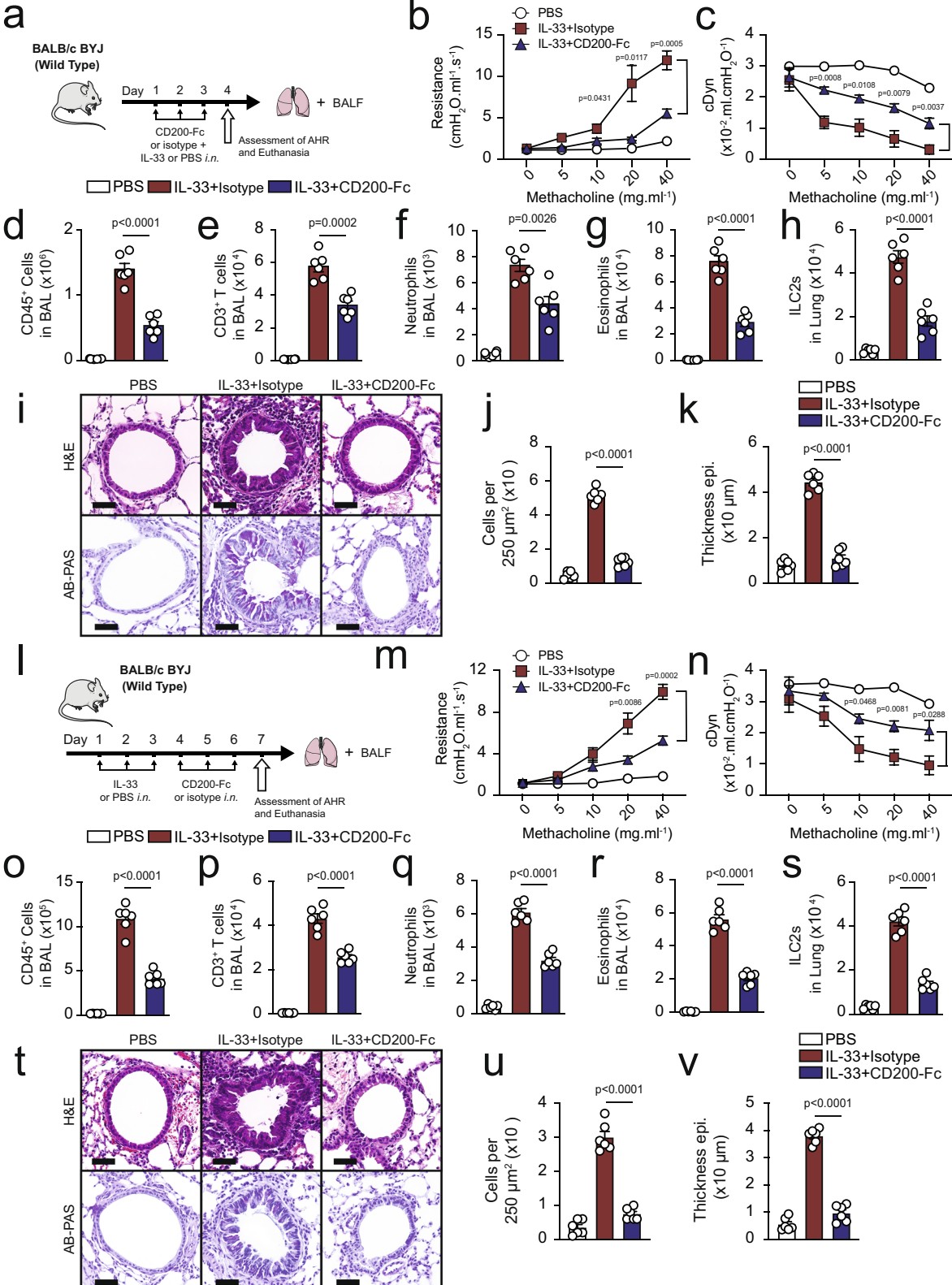

The relevant cytokine and cytokine receptor genes were displayed on the heatmap based on their modulation in response to CD200-Fc treatment (Fig. 5d). Downregulation of ILC2 relevant proinflammatory cytokines, such as *Il5*, *Il6*, *Il9*, and *Csf2* (GM-CSF) and upregulation of anti-inflammatory *Il10* at a transcriptional level is consistent with our previous ex vivo data (Fig. 1g).

Consistent with our previous findings (Fig. 2e, f), CD200R engagement reduced the expression of transcriptional factors *Gata3*. CD200R inhibition of ILC2s also modulated the gene expression of other transcriptional factors, such as *Rela*, *Nfkbia*, and *Nfkb2* (Fig. 5e). Falling in line with these results, scRNAseq profiles of 4428 IL-33-activated ILC2s isolated from WT mice suggested relatively reduced expression of *Nfkb1* (Fig. 5f) and

**Fig. 3 CD200R engagement ameliorates IL-33-induced AHR and pulmonary inflammation. a** A cohort of WT were challenged with rmIL-33 (0.5 μg) or PBS intranasally (i.n.) and treated with CD200-Fc or isotype control on days 1, 2, and 3, $n = 6$ mice. On day 4, we assessed the lung function and acquired the samples, as shown in the timeline. **b, c** Line graph show lung resistance and dynamic compliance (cDyn) in response to increasing doses of methacholine. Total numbers of CD45$^+$ cells (**d**), CD3$^+$ T cells (**e**), neutrophils (**f**), and eosinophils (**g**) in BAL fluid have been demonstrated in the bar graphs. **h** the numbers of ILC2s in the lungs. **i** representative images, and **j, k** quantification of hematoxylin and eosin (H&E) and alcian blue/periodic acid–schiff (AB-PAS) stained histologic sections of the lungs of mice. Scale bars = 50 μm. **l** A cohort of WT were challenged with rmIL-33 (0.5 μg) or PBS intranasally (i.n.) on days 1–3, $n = 6$ mice. Subsequently, the mice treated intranasally with CD200-Fc or isotype control for three days. On day 7 the lung function was assessed, and samples were collected, as shown in the timeline. **m** lung resistance. **n** dynamic compliance. Bar graphs displaying the total numbers of CD45$^+$ cells (**o**), CD3$^+$ T cells (**p**), neutrophils (**q**), and eosinophils (**r**) in BAL fluid. **s** Number of ILC2s in lungs. **t** Representative images and quantification (**u, v**) of H&E and AB-PAS stained histologic sections of the lungs of mice. Scale bars = 50 μm. Data are shown as means ± SEMs and are representative of three individual experiments. Statistical analysis, one-way ANOVA and two-tailed student's $t$-test. Mouse and lung images are sourced through an open access license from Servier Medical Art.

*Nfkb2* (Fig. 5g). In order to confirm our RNA-seq results, we examined the protein expression levels of pIKKα/β, as well as p65 and p52 in the canonical and non-canonical NF-κB pathways, respectively. Activated ILC2s were isolated from lungs of WT mice after three doses of i.n. IL-33 and ex vivo treated with CD200-Fc or isotype control. After 24 h, intracellular pIKKα/β, p52, and p65 (phosphorylated forms) were analyzed by flow cytometry. In agreement with our transcriptional findings, pIKKα/β (Fig. 5h), p65 (Fig. 5k), and p52 (Fig. 5l) were all downregulated at a protein level by CD200R stimulation. *Rela* (Fig. 5i) and *Relb* (Fig. 5j) were similarly downregulated at a mRNA level by CD200R agonistic treatment. With these findings, CD200R engagement serves as a potent inhibitory signal for secretion of type 2 cytokines, and inhibits both the canonical and non-canonical NF-κB signaling pathway via pIKKα/β, p52, and p65 in activated pulmonary ILC2s.

**CD200R is inducible on human ILC2s and inhibits cytokine production.** In order to explore the translational relevance of our findings, we next examined the dominant sources of CD200 within the human lung by analyzing whole tissue scRNA sequencing data of healthy donors and asthmatic patients, as described before[17]. Our results demonstrate there are multiple sources of CD200 within human lung tissue (Fig. 6a). Smooth muscle cells, endothelial cells, submucosal cells, ciliated, and mucous ciliated cells all appear to be the major sources of CD200 in the lungs of both healthy individuals and asthmatics. Interestingly, CD200 appeared to be upregulated in at least eight cells types (highlighted in bold), including fibroblasts and T cells in asthmatic patients. Next, we continued to investigate whether human ILC2s express CD200R. Using human PBMCs of healthy donors, we FACS purified ILC2s on the basis of the lack of expression of human lineage markers and expression of CD45, CRTH2, and CD127 (Fig. 6b) and then cultured them with recombinant human (rh) IL-2 and rhIL-7. Additionally, the human ILC2s were ex vivo stimulated with rhIL-33, and the expression of CD200R was analyzed over time at 0, 24, 48, and 72 h by flow cytometry (Fig. 6c). We discovered that CD200R is expressed on both naïve and activated human ILC2s. We also observed that CD200R expression is inducible by rhIL-33 and first reached statistical significance after 24 h of ex vivo rhIL-33 stimulation (Fig. 6d). We next inquired whether CD200R engagement can inhibit the effector function of activated human ILC2s. Purified peripheral blood ILC2s from healthy donors were cultured with rhIL-2, rhIL-7, rhIL-33, and ex vivo stimulated with human specific CD200-Fc (10 μg/mL) or isotype control. After 48 h, the supernatant was collected, and cytokine levels were measured via Luminex assay. Consistent with our mouse ILC2 observations, IL-4, IL-5, IL-6, IL-9, IL-13, and GM-CSF levels were inhibited by CD200R engagement on activated human

ILC2s (Fig. 6e). With these results, we are confident in the potential that CD200R engagement may have in cases of ILC2-mediated human diseases.

**CD200R engagement on human ILC2s ameliorates IL-33-induced AHR.** Pushing for more translational potential, we explored the efficacy of anti-CD200R agonistic treatment in preventing human ILC2-mediated AHR. We purified ILC2s from human PBMCs of healthy donors and, after 48 h of ex vivo culture in the presence of rhIL-2 and rhIL-7, adoptively transferred the cells through the tail vein to $Rag2^{-/-}Il2rg^{-/-}$ mice that lack T cells, B cells, and NK cells and ILCs. Subsequently, the humanized mice were challenged with 1 μg of rhIL-33 or PBS intranasally and treated with either human specific CD200-Fc or isotype control i.n. on days 1, 2, and 3 (Fig. 7a). We measured lung function and assessed the BAL fluid by flow cytometry on the fourth day. Lung function analysis demonstrated that intranasal rhIL-33 administration increased lung resistance (Fig. 7b) and decreased dynamic compliance (Fig. 7c) in isotype treated mice, but not in CD200-Fc-treated cohort. Challenge with rhIL-33 led to increased eosinophilia (Fig. 7d) and increased human ILC2 numbers (Fig. 7e) in the recipients of isotype control, when compared to CD200-Fc treated group. Histological analysis confirmed these findings, (Fig. 7f) revealing significantly less presence of inflammatory cells (Fig. 7g), and reduced thickness of the epithelium (Fig. 7h) in CD200-Fc-treated cohort. Summarized, CD200R stimulation of human ILC2s can prevent AHR in humanized IL-33 mouse model.

In order to determine the therapeutic efficacy of anti-CD200R treatment in reversing established AHR driven by human ILC2s, we next purified human ILC2s, ex vivo stimulated them with rhIL-2 and rhIL-7 for 48 h, and adoptively transferred the cells through the tail vein to $Rag2^{-/-}Il2rg^{-/-}$ recipients. We challenged the humanized mice for three consecutive days, and subsequently treated them with either human specific CD200-Fc or isotype control for three additional days. The lung function was measured, and samples were collected on the seventh day (Fig. 7i). In agreement with the preventative model, CD200R engagement on human ILC2s reduced lung resistance (Fig. 7j) and significantly improve dynamic compliance (Fig. 7k). Treatment with CD200-Fc reduced abrogated inflammatory eosinophil recruitment, and thus reduced eosinophilia in BAL fluid (Fig. 7l) and reduced the number of human ILC2s in the lungs (Fig. 7m). The histological analysis of the lungs (Fig. 7n) further confirmed reduced number of inflammatory cells (Fig. 7o) and thickening of the epithelium (Fig. 7p). Our results indicate that CD200R engagement on human ILC2s in vivo can potentially serve as a viable therapeutic option for amelioration of human ILC2-mediated asthma and other lung inflammatory diseases.

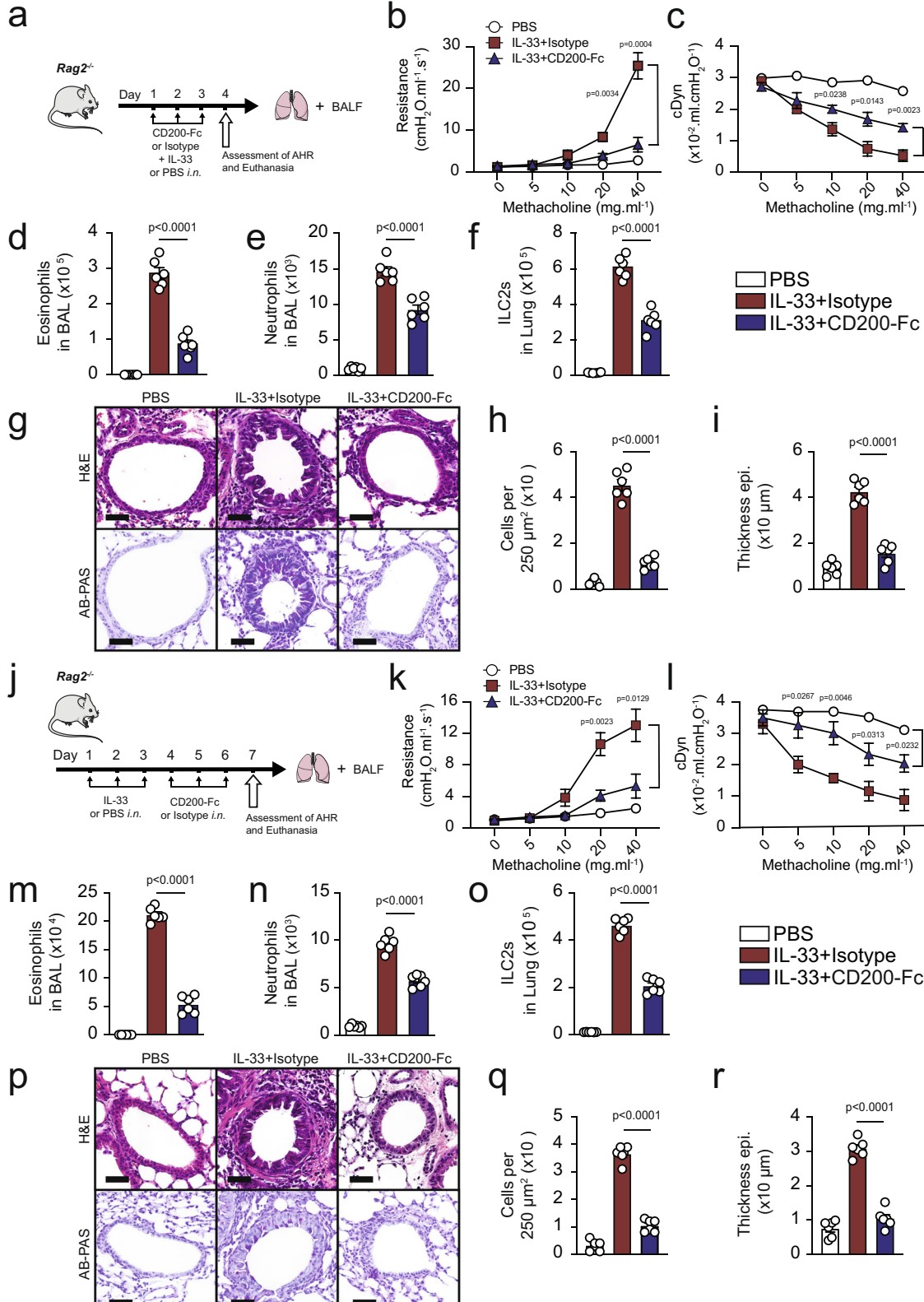

## Discussion

In the present study, we elucidate the mechanism, signaling pathways, and therapeutic potential of CD200R engagement on pulmonary ILC2s in context of allergic asthma, AHR, and lung inflammation. We ascertain that CD200R, a cell surface inhibitory checkpoint, is expressed on both human and mouse ILC2s, and this expression is inducible by IL-33. CD200R expression pattern at the mRNA and protein levels appeared distinct, which may be explained by the reported differences in transcription/translation kinetics, as well as the reduced longevity of mRNA compared to protein[18–20]. The IL-33/ST2-dependent induction of CD200R on activated ILC2s during pulmonary inflammation underscores the therapeutic viability of targeting this regulatory axis, and contributes to the high efficacy of anti-CD200R

**Fig. 4 CD200R engagement ameliorates ILC2-dependent AHR. a** A cohort of $Rag2^{-/-}$ were challenged with rmIL-33 (0.5 μg) or PBS intranasally (i.n.) and treated with CD200-Fc or isotype control on for three days, $n = 6$ mice. On day 4, we assessed the lung function and acquired the samples, as shown in the timeline. Lung resistance (**b**) and dynamic compliance (**c**) in response to increasing doses of methacholine. Total numbers of eosinophils (**d**) and neutrophils (**e**) in BAL fluid and ILC2s in the lungs (**f**) are demonstrated in the bar graphs. **g** representative histological images, **h, i** quantification of H&E and AB-PAS stained histologic sections of the lungs. Scale bars = 50 μm. **j** A cohort of $Rag2^{-/-}$ mice were challenged with rmIL-33 (0.5 μg) or PBS intranasally (i.n.) for three days, $n = 6$ mice. Subsequently, the mice were treated intranasally with CD200-Fc or isotype control for an additional three days. On day 7 the lung function was assessed, and samples were acquired. **k** Lung resistance. **l** dynamic compliance. Bar graphs displaying the numbers of BAL eosinophils (**m**), neutrophils (**n**), and pulmonary ILC2s (**o**). **p** Representative images and quantification (**q, r**) of H&E and AB-PAS stained histologic sections of the lungs of mice. Scale bars = 50 μm. Data are shown as means ± SEMs and are representative of three individual experiments. Statistical analysis, one-way ANOVA and two-tailed student's $t$-test. Mouse and lung images are sourced through an open access license from Servier Medical Art.

treatment in modulating effector function of activated ILC2s and alleviating AHR. Interestingly, CD200R is also expressed on naïve ILC2s, which could denote a potential role in maintaining naïve ILC2 homeostasis. Since naïve ILC2s do not have a measurable effector function, future studies and novel assays are needed to explore the role of inhibitory receptors, such as CD200R on naïve ILC2s. We observed that CD200R engagement modulates activated ILC2 effector function and significantly inhibits type 2 cytokine production in both mouse and human ILC2s. Further supporting a potent immunoregulatory role on ILC2s, CD200R reduces ILC2 activation and proliferation. Although CD200R did not cause polarization towards either ILC1 or ILC3 phenotypes, we did observe induction of Blimp-1 by CD200R engagement. Since our group recently reported ILC2$_{10}$s are mainly controlled by transcription factors Blimp-1 and cMaf[21], these findings suggest IL-10 production via CD200R engagement leverages the Blimp-1 pathway and partially mimics mouse ILC2$_{10}$s. CD200R engagement did not induce IL-10 in human ILC2s, suggesting induction of IL-10 in humans may require different signaling pathways that warrant additional studies. Moreover, mouse ILC2s in this study were isolated from lung but the human ILC2s were isolated from PBMCs; therefore, the tissue specific signature of ILC2 may also contribute to production of cytokines such as IL-10. We further demonstrate that CD200R engagement inhibits both the canonical and non-canonical NF-κB pathways in activated ILC2s, as evidenced by downregulation of pIKKα/β, $Nfkb1$, and $Rela$ (p65), as well as $Nfkb2$ (p52) and $Relb$. Using a CD200-Fc, we showed that CD200R engagement on lung-derived ILC2s both protects against development of asthma, and is capable of therapeutically reversing established AHR in various mouse models.

Initially, we utilized an IL-33-based model of AHR because both IL-33 and IL-25 have been previously shown to induce ILC2-mediated AHR and lung inflammation, although IL-33 is more potent than IL-25[22]. In order to exclude the effects of adaptive immune system cells in amelioration of AHR, we then utilized $Rag2^{-/-}$ mice that lack mature B and T cells[23]. Since alarmins, such as IL-33, are not naturally found in the environment, we subsequently examined the effect of CD200R agonistic treatment on lung inflammation and AHR induced by clinically relevant allergens. We utilized *Alternaria alternata* because it is commonly found in the environment and is a well-known allergen in humans[24]. *Alternaria alternata* has been reported to cause allergic inflammation in mice independent of adaptive immunity, making it an ideal model to study ILC2-dependent asthma[25,26]. Our result show CD200R engagement on all ILC2s improves lung function and curtails lung inflammation following exposure to principal allergenic agents.

Understanding the critical interplay between ILC2s and their respective physical tissue niches during homeostasis, and/or inflammation remains an area of immense interest[27,28]. Near the pulmonary epithelial interface, previous studies have highlighted the importance of CD200–CD200R axis in fine tuning the

immunological tolerance, through modulation of dendritic cells and pulmonary alveolar macrophages[9,29,30]. We demonstrate CD200R expression on macrophages and ILC2s is comparable at steady state in the lungs. The ligand of CD200R, CD200 has no known signaling motif and is reportedly expressed by the epithelial cells within the lungs[12–14]. Since pulmonary ILC2s are located subjacent to the epithelium at less than 70 μm away from the bronchioles, they can potentially interact with adjacent CD200-expressing epithelial cells[7]. Additionally, various T cells have been shown to express CD200, and CD200 upregulation has been associated with Treg expansion[31]. Our group previously has shown that Treg directly interact with, and suppress ILC2s via an ICOS:ICOS-Ligand dependent mechanism[32]. Fibroblast-like adventitial stromal cells (ASCs) have also been shown to modulate local ILC2 effector function and expansion via IL-33 and TSLP secretion[33]. Interestingly, we found CD200 ligand is upregulated in at least eight pulmonary cells types, including fibroblasts and T cells in patients with asthma. We further report multiple sources of CD200 ligand in the lungs of both healthy individuals and asthmatics, including smooth muscle cells, endothelial cells, submucosal cells, ciliated, and mucous ciliated cells. We believe the CD200–CD200R axis exemplifies a scenario in which these local cells in the lung provide an inhibitory signal for ILC2s suppression most probably via a contact-dependent mechanism.

We explored the influence of CD200R dependent stimulation of human ILC2s in context of asthma, and demonstrated that expression of CD200R is inducible on human ILC2s by IL-33 in a time-dependent manner. Consistent with our mouse results, CD200R engagement inhibits type 2 cytokine production in activated human ILC2s. In order to further assess the clinical relevance of our findings, we assessed the effect of anti-CD200R treatment in preventative and therapeutic humanized mouse models in which human peripheral ILC2s are adoptively transferred to $Rag2^{-/-}Il2rg^{-/-}$ mice, followed by intranasal administration of recombinant human IL-33 to induce AHR. These humanized mouse models provide a unique platform for investigating the contribution of immune cells to human asthma, and enable the assessment of therapeutic targets in preclinical studies. Human IL-5 has been shown to activate mouse eosinophils, highlighting the viability of using humanized mice in eosinophilic inflammatory studies[34,35]. Our results emphasize the critical role of CD200R as an immunoregulatory checkpoint of ILC2s, provide insights into the role of CD200R in pulmonary ILC2s and anti-CD200R treatment as a promising therapeutic avenue to prevent, and treat allergic asthma and pulmonary inflammation. Additionally, our humanized mice experiments utilized a human CD200-Fc, which specifically binds and engages human CD200R[12,36,37]. Therefore, these results further demonstrate in vivo CD200R engagement on human ILC2s is sufficient for amelioration of ILC2-driven AHR and reduction of eosinophilia.

Overall, our results suggest that the CD200–CD200R axis provides an important inhibitory signal for the modulation of

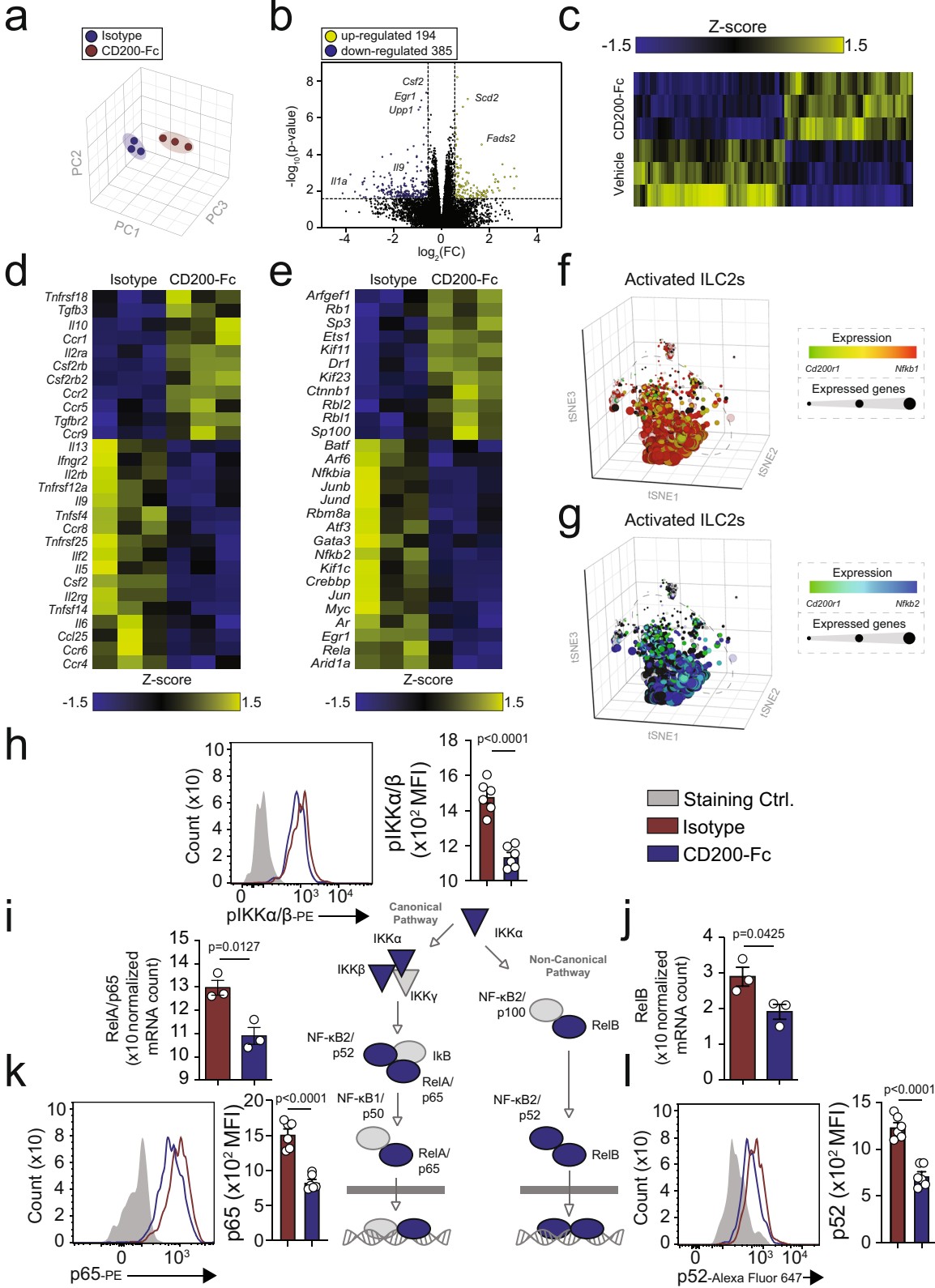

human ILC2s. In clinical settings, asthmatic patients have an established AHR. Therefore, it is important to not only prevent, but also to reverse their established pulmonary inflammation. Our humanized mouse models demonstrate that CD200R engagement asserts not only a preventive role against the onset of AHR, but also a therapeutic effect in models with previously established asthma. We introduce CD200R-targeted agonistic

treatment as a promising therapeutic avenue to reverse AHR in asthmatic patients

## Methods
**Mice.** Wild-type (WT) BALB/cByJ, recombination-activating gene 2-deficient (C.B6(Cg)-Rag2tm1.1Cgn/J), recombination-activating gene 2-deficient gamma-chain-deficient (C;129S4-Rag2tm1.1Flv Il2rgtm1.1Flv/J) mice were purchased from

**Fig. 5 CD200R signaling inhibits the canonical/non-canonical NF-κB pathways.** In vivo activated lung-derived mouse ILC2s were cultured in presence of recombinant mouse (rm)IL-2, rmIL-7, and rmIL-33 and stimulated CD200-Fc (10 μg/mL) or isotype control for 24 h. Total RNA was isolated and sequenced. **a** Principal Component Analysis (PCA) of activated WT ILC2s treated with CD200-Fc (red) or isotype control (blue). **b** Volcano plot comparison representing whole transcriptome gene expression of sorted WT ILC2s treated with either isotype control or CD200-Fc for 24 h ex vivo. Differentially expressed genes (described as statistically significant adjusted *p*-value ≤ 0.05; GSA statistics using lognormal with shrinkage) with changes of at least 1.5-fold change (FC) are shown in yellow (upregulated) and blue (downregulated). Relevant differentially expressed genes are identified. **c** Heat plot of all differentially expressed genes, **d** selected cytokine, and cytokine receptor genes, and **e** transcription factors. **f**, **g** scRNAseq profiles of 4428 IL-33-activated ILC2s from WT mice. Cells are colored based on expression levels of *Cd200r1, Nfkb1, and Nfkb2*. Dot size is indicative of the total gene expression level of each cell. Dashed lines enclose cells with highest CD200R expression. Representative protein expression of pIKKα/β (**h**), mRNA expression of RelA/p65 and RelB (**i**, **j**). The representative protein expression of NF-κB p65 (**k**), and NF-κB p52 (**l**) in isolated ILC2s from WT mice challenged with IL-33 and cultured ex vivo for 24 h with CD200-Fc (red) or isotype control (blue), *n* = 6 mice. The staining FMO controls are shown as gray. The corresponding quantifications are presented as mean fluorescence intensity (MFI), and the error bars denote the mean ± SEM and are representative of three individual experiments. Statistical analysis, two-tailed student's *t*-test.

the Jackson Laboratory (Bar Harbor, Me), and bred in our animal facility at the University of Southern California. All mice were from a BALB/cByJ background. Mice were maintained at macroenvironmental temperature of 21–22 °C, humidity (48–52%), in a conventional 12:12 light/dark cycle with lights on at 6:00 a.m. and off at 6:00 p.m. We used 5-week-old to 8-week-old age-matched female mice in our studies. All experimentation protocols were approved by the USC Institutional Animal Care and Use Committee and conducted in accordance with the principles of the Declaration of Helsinki.

**Induction and measurement of airway hyperreactivity**. The mice were intranasally challenged via rmIL-33 (BioLegend, San Diego, Calif) or *Alternaria alternata* extracts (Greer Laboratories, Lenoir, North Carolina), as shown in the experimental schemes. For CD200R engagement, mouse CD200-Fc (3355-CD-050; R&D Systems), human CD200-Fc (2724-CD-050; R&D Systems) and the corresponding IgG1 isotypes (110-HG-100; R&D Systems) were used. Lung function was measured by direct measurement of lung resistance and dynamic compliance in anesthetized tracheostomized mice, in which mice were mechanically ventilated via the FinePointe RC system (Buxco Research Systems, Wilmington, NC), and sequentially challenged with aerosolized increasing doses of methacholine[38]. Maximum lung resistance and minimum compliance values were recorded during a 3-min period after each methacholine challenge. AHR data were analyzed by repeated measurements of a general linear model.

**Collection and preparation of bronchoalveolar lavage fluid**. After measurements of AHR, the trachea was cannulated and the bronchial alveolar lavage (BAL) fluid was collected as described before[39]. Briefly, we tracheostomized and intubated the mice and then washed the airways three times with 1 mL of ice-cold PBS each time, followed by centrifuging at 400 × g for 7 min and harvesting the cells. Data were analyzed with FlowJo software (TreeStar, Ashland, Ore). The absolute cell numbers in BAL fluid were calculated by means of flow cytometry by staining the cells with phycoerythrin (PE)–anti-Siglec-F (E50-2440; BD Biosciences, San Jose, Calif; 1/1000), fluorescein isothiocyanate(FITC)–anti-CD19 (6D5; 1/400), peridinin-chlorophyll-protein complex (PerCP)/Cy5.5–anti-CD3ε (17A2; 1/200), allophycocyanin (APC)–anti–Gr-1 (RB6-8C5; 1/1000), PE/Cy7–anti-CD45 (30-F11; 1/500), APC/Cy7–anti-CD11c (N418; BioLegend, San Diego, Calif; 1/400), and eFluor450–anti-CD11b (M1/70; eBioscience, San Diego, Calif; 1/500) in the presence of anti-mouse FC-block (2.4G2; BioXcell, West Lebanon, NH; 1/200). We used CountBright Absolute Count Beads (Thermo Fisher Scientific, Waltham, Mass), according to the manufacturer's instructions. At least $10^5$ CD45$^+$ cells were acquired on a BD FACSCanto II (BD Biosciences) using the BD FACSDiva software v8.0.1.

**Lung tissue preparation for flow cytometry**. Utilizing fine surgical scissors, mouse lungs were surgically removed and minced in a sterile environment subsequently incubated in type IV collagenase (1.6 mg/mL; Worthington Biochemicals, Lakewood, NJ) at 37 °C for 60 min. After digestion, mouse lung fragments were then pressed through a 70 μm nylon cell strainer, using the rubber end of a sterile 10 mL syringe plunger, in order to create a single cell suspension. In order to terminate the enzymatic reaction of collagenase, the cells were washed with 1× phosphate buffered saline (PBS) by centrifugation at 400 × g for 7 min at 4 °C. In order to exclude and lyse the red blood cells (RBCs), the cell pellet was subsequently resuspended in 1× RBC lysis buffer (Biolegend®, San Diego, CA) and incubated at room temperature (RT) for 5 min. In order to terminate the chemical reaction, the cells were subsequently washed and centrifuged—at 400 × g for 7 min at 4 °C—with 1× PBS. The remaining pellet was then further prepared for flow cytometry[40]. The absolute cell numbers in lung tissue were calculated by means of flow cytometry. Mouse ILC2s were defined based on the lack of expression of classical lineage markers (CD3e, CD5, CD45R, Gr-1, DX5, CD11c, CD11b, Ter119, NK1.1, TCR-γδ, and FCεRI) and expression of CD45, ST2, and CD127. All ILC2s were stained with Biotinylated anti-mouse lineage (CD3e (145-2C11; 1/200), CD5

(53-7.3; 1/200), CD45R (RA3-3B2; 1/200), Gr-1 (RB6-8C5; 1/200), DX5 (DX5; 1/200), CD11c (N418; 1/200), CD11b (M1/70; 1/200), Ter119 (TER-119; 1/200), NK1.1 (PK136; 1/200), TCR-b (H57 597; 1/200), TCR-γδ (GL3; 1/200), and FcεRIa (MAR-1; 1/200)), FITC–anti-streptavidin (405202; BioLegend; 1/500), phycoerythrin (PE)–anti-CD200R (181; SinoBiologica; 1/200), APC/Cy7–anti-CD45 (30-F11; 1/200), PE/Cy7–anti-CD127 (A7R34; BioLegend, San Diego, Calif; 1/200), peridinin–chlorophyll–protein complex (PerCP)/Cy5.5–anti-ST2 (RMST2-2; eBioscience; 1/200). Mouse macrophages were defined and stained based on expression of BV650–CD45 (30-F11; 1/200), BV421–CD11b (M1/70; 1/200), and PE/Cy7–F4/80 (BM8; 1/200). We used CountBright Absolute Count Beads (Thermo Fisher Scientific, Waltham, Mass), according to the manufacturer's instructions. At least $10^5$ CD45$^+$ cells were acquired on a BD FACSCanto II (BD Biosciences). Intranuclear staining with APC anti-mouse Ki67 (SolA15, eBioscience; 1/50), PE anti-mouse GATA3 (TWAJ; Invitrogen; 1/50), APC anti-mouse T-bet (4b10, Thermofisher; 1/50), PE anti-mouse RORγt (AFKJS-9, Thermofisher; 1/50), PE anti-mouse FOXP3 (MF-14, BioLegend; 1/50), PE anti-mouse cMaf (sym0F1, Thermofisher; 1/50), and APC anti-mouse Blimp-1 (5E7, BioLegend; 1/50) were performed using the Foxp3 Transcription Factor Staining Kit (ThermoFisher Scientific)[41]. Alexa 647 anti-mouse NFκB p52 (C5; 1/50) was purchased from Santa Cruz Biotechnology. PE anti-human/mouse RelA NFκB p65 (IC5078P; 1/50) and PE IgG2B (133303; 1/50) was purchased from R&D Systems. PE phospho-IKKα/β (Ser176/180) (16A6; 1/50) Rabbit mAb was obtained from Cell Signaling Technologies. Intracellular staining was performed to assess cytokine production using PE anti-mouse/human IL-5 (TRFK5; BioLegend; 1/50) and eFluor450 anti-mouse IL-13 (eBio13A; eBioscience; 1/50) after 4 h of in vitro incubation with 50 μg/mL PMA (Sigma), 500 μg/mL ionomycin (Sigma), and 1 μg/mL Golgi plug. Staining controls PE IgG XP (DA1E, Cell Signaling Technology, 1/50), eFluor 450-IgG1 kappa (eBRG1, Thermofisher, 1/50), APC IgG2a kappa (eBR2a, Thermofisher, 1/50), PE Rabbit Hamster IgG (60024B, R&D Systems, 1/50) were utilized. CountBright Absolute Count Beads were used to count lung immune cells (Invitrogen). Acquisition was performed on a BD FACSCanto II (BD Biosciences) using the BD FACSDiva software v8.0.1. Data were analyzed with FlowJo software (TreeStar) version 10.

**ILC2 sorting and ex vivo stimulation**. All cells were purified by flow cytometry using BD FACS ARIA III (BD biosciences, San Jose, CA) with a purity of >95%. The isolated ILC2s were cultured (at least $5 × 10^3$ cells/well) in 96-well round-bottom plates with Gibco™ Roswell Park Memorial Institute (RPMI) 1640 medium (Thermo Fisher Scientific, Waltham, MA) that was supplemented with 10% fetal bovine serum (FBS), 2% antibiotics (penicillin and streptomycin), and 0.05 mM b-mercaptoethanol. The cells were maintained in a 37 °C incubator with 5% CO$_2$. All mouse ILC2s were cultured in the presence of recombinant mouse (rm) IL-2 (10 ng/mL), rmIL-7 (10 ng/mL), and/or rmIL-33 (10 ng/mL). The mouse cells were stimulated via mouse CD200-Fc (2724-CD-050; R&D Systems) and the corresponding IgG1 isotypes (110-HG-100; R&D Systems). In some experiments, cells were treated with either anti-ST2 blocking antibody (DIH4; BioLegend) or IgG1, κ isotype control (RTK2071; BioLegend). Human ILC2s were cultured in the presence of rhIL-2 (10 ng/mL), rhIL-7 (10 ng/mL), and/or rhIL-33 (10 ng/mL). The human cells were stimulated via human CD200-Fc (2724-CD-050; R&D Systems) and the corresponding IgG1 isotypes (110-HG-100; R&D Systems).

**Human ILC2 isolation and adoptive transfer**. All human studies were approved by USC Institutional review board and conducted in accordance with the principles of the Declaration of Helsinki. Informed consent was obtained from all human participants. Human blood samples were obtained from male and female healthy donors (age 18–65). Human peripheral blood ILC2s were isolated from total PBMCs as described previously[42]. Briefly, human fresh blood was first diluted 1:1 in PBS and transferred to SepMateTM-50 separation tubes (STEMCELL Technologies) prefilled with 15 mL LymphoprepTM (Axis-Shield). Samples were centrifuged at 1200 × g for 10 min to collect PBMCs. The CRTH2 MicroBead kit

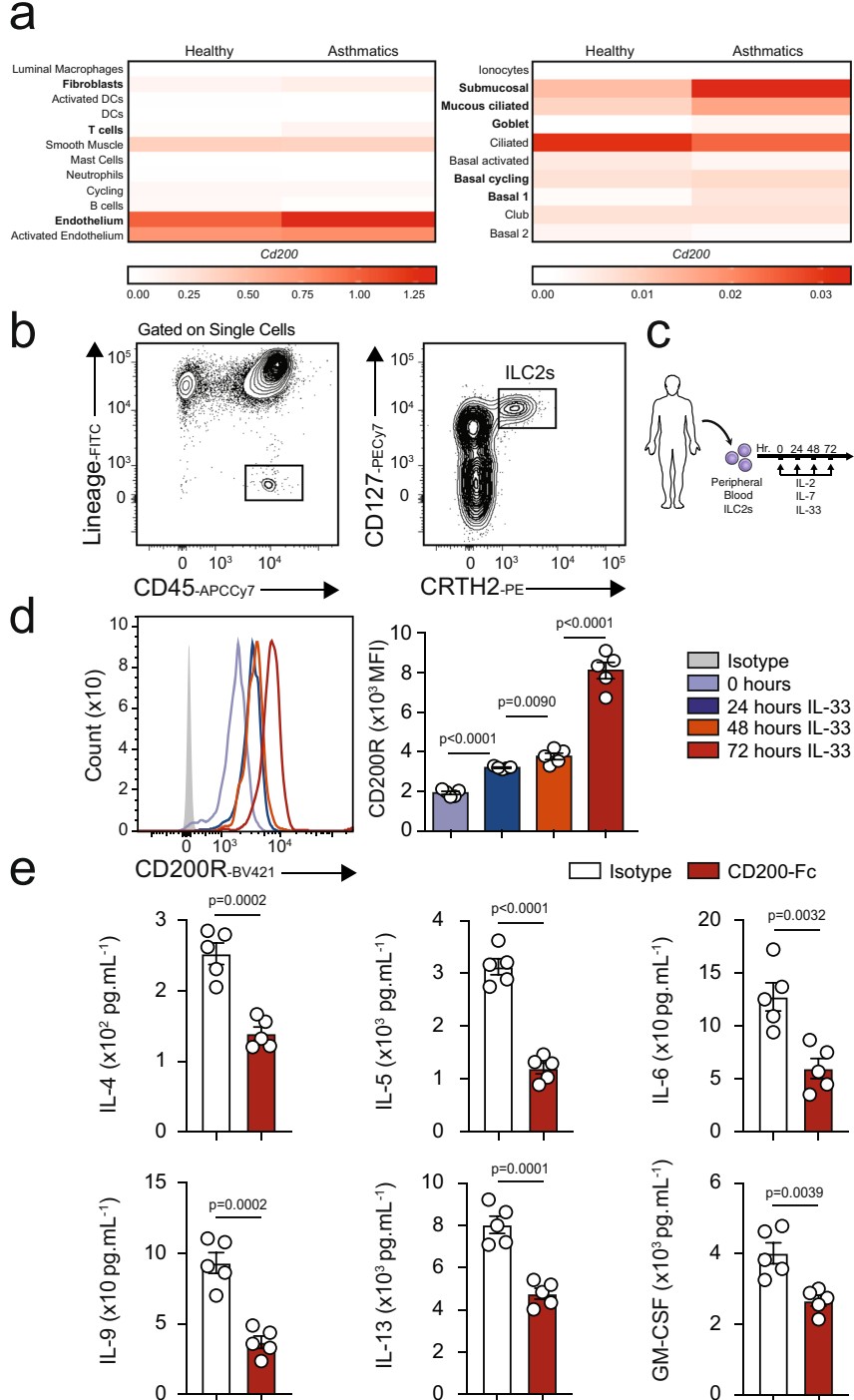

**Fig. 6 Human ILC2s express CD200R and this expression is inducible by IL-33. a** CD200 mRNA expression among 22 cell types in lungs of healthy donors and asthmatic patients. **b** The gating strategy of human ILC2s identified as Lin⁻CD45⁺CD127⁺CRTH2⁺ cells. **c** Human peripheral-blood ILC2s were freshly sorted and cultured with 10 ng of recombinant human (rh)IL-2 and rhIL-7 in presence or absence of rhIL-33 (10 ng) for 24, 48, and 72 h. Freshly isolated ILC2 at 0 h and ex vivo activated ILC2s were analyzed by flow cytometry as indicated in the scheme. **d** kinetics of CD200R induction by IL-33. **e** The levels of IL-4, IL-5, IL-6, IL-9, IL-13, and GM-CSF in the culture supernatants were measured by Luminex after 48 h of stimulation with CD200-Fc (10 μg/mL). Data representative of five individual blood donors, $n = 5$. Error bars are the mean ± SEM. Statistical analysis, one-way ANOVA and two-tailed student's $t$-test. Human and cell images are sourced through an open access license from Servier Medical Art.

(Miltenyi Biotec) was then used according to the manufacturer's conditions in order to isolate CRTH2+ cells. Human ILC2s were sorted from CRTH2+ cells based on the lack of expression of classical lineage markers and expression of CD45, CRTH2, and CD127. The following human antibodies were used: FITC-Lineage that includes mixture of CD3, CD14, CD16, CD19, CD20, CD56 (348801; 1/100), FITC-CD235a (HI264; 1/500), FITC-FCeRIa (AER-37; 1/100), FITC-CD1a

(HI149;1/100), FITC-CD123 (6H6; 1/100), FITC-CD5 (L17F12; 1/100), APCCy7-CD45 (HI30; 1/100), PE-CD294 (CRTH2) (BM16; 1/100), PE/Cy7-CD127 (A019D5; 1/100), and BV421 anti-CD200R (OX-108; 1/200) were purchased from BioLegend. Purified human ILC2s were cultured with rhIL-2 (20 ng/mL) and rhIL-7 (20 ng/mL) and adoptively transferred to $Rag2^{-/-} Il2rg^{-/-}$ mice (5 × 10⁴ cells per mouse)[43].

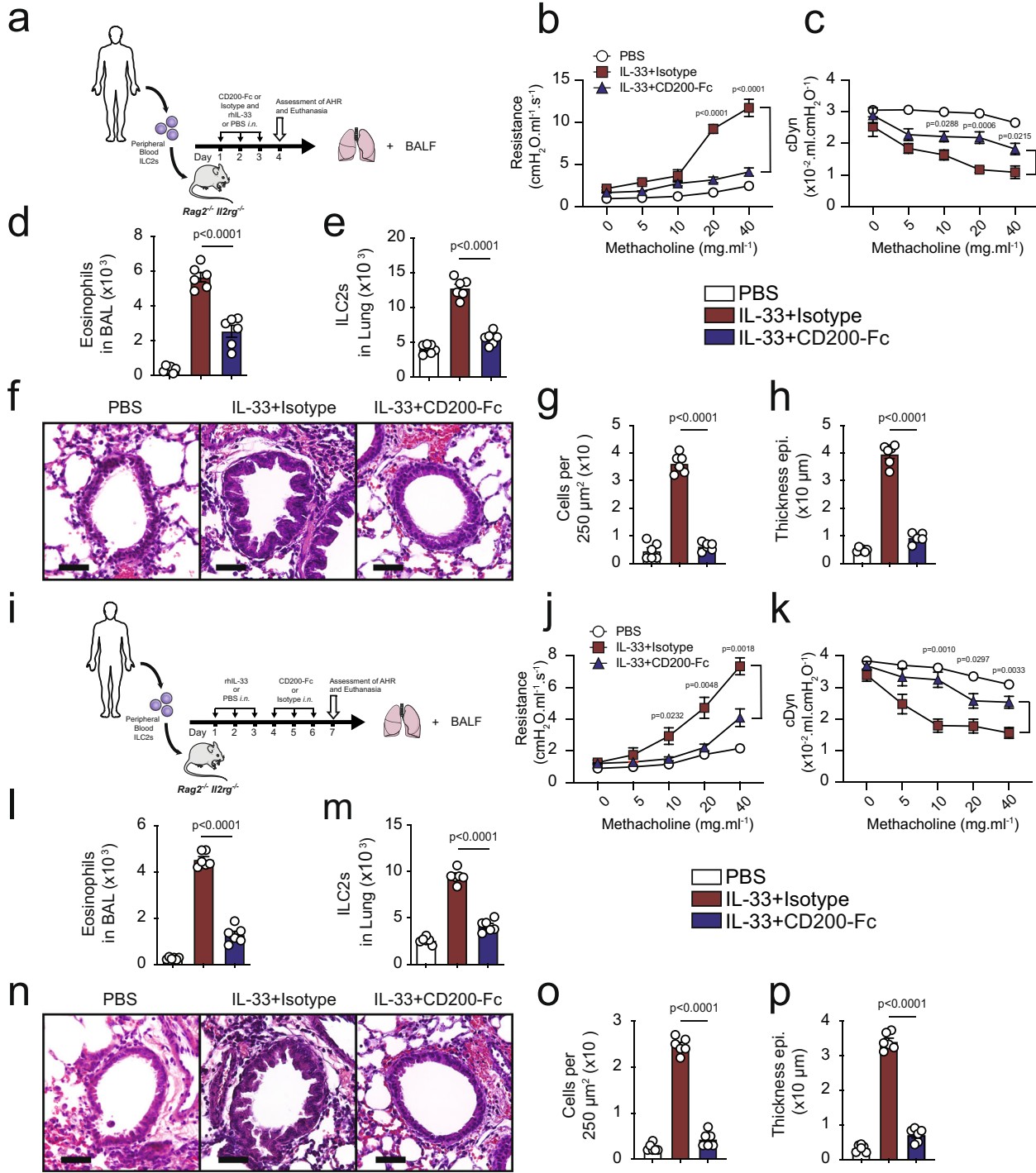

**Fig. 7 CD200R engagement ameliorates human ILC2-mediated AHR. a** Human peripheral ILC2s from healthy donors were purified via FACS and cultured with rhIL-2 (20 ng/ml) and rhIL-7 (20 ng/ml) for 48 h, and then adoptively transferred into $Rag2^{-/-}Il2rg^{-/-}$ mice. Additionally, the mice were challenged with either rhIL-33 (1 μg) or PBS intranasally (i.n.) and treated with CD200-Fc or isotype control on days 1, 2, and 3. Measurement of lung function and analysis of BAL fluid followed on day 4, as shown in the timeline, $n = 6$ mice. **b**, **c** Line graphs show lung resistance and dynamic compliance in response to increasing doses of methacholine. **d** The numbers of eosinophils in the BAL. **e** The numbers of human ILC2s in the lungs. **f** Histological images of the lungs, and **g** corresponding quantifications of cell numbers, and **h** thickness of epithelium. Scale bars = 50 μm. **i** Human peripheral ILC2s from healthy donors were purified via FACS and cultured with rhIL-2 (20 ng/ml) and rhIL-7 (20 ng/ml) for 48 h, and then adoptively transferred into $Rag2^{-/-}Il2rg^{-/-}$ mice. The mice were then challenged with either rhIL-33 (1 μg) or PBS intranasally (i.n.) on days 1, 2, and 3. Subsequently, the mice were treated with CD200-Fc or vehicle control on days 4, 5, and 6. The lung function, BAL fluid and lung tissue were analyzed on day 7, $n = 6$ mice. **j** Lung resistance, and **k** dynamic compliance. **l** BAL eosinophilia and **m** pulmonary ILC2s are shown as bar graphs. **n** Histological images of the lungs, and **o**, **p** corresponding quantifications. Scale bars = 50 μm. Data are shown as means ± SEMs and are representative of three individual experiments. Statistical analysis, one-way ANOVA followed by Tukey post-hoc tests. Human, mouse and lung images are sourced through an open access license from Servier Medical Art.

**RNA-sequencing and data analysis**. Freshly sorted ILC2s after three intranasal administrations of rmIL-33 (0.5 μg per mouse), defined in this study as IL-33-activated ILC2s (aILC2s), were incubated with rmIL-2 (10 ng/mL) and rmIL-7 (10 ng/mL) with CD200-Fc or the isotype control for 24 h ($5 \times 10^4$ mL$^{-1}$). Total RNA was isolated using MicroRNAeasy (Qiagen). scRNA sequencing data [GSE102299] for mouse and human samples were obtained as described before[17,44]. In total, 10 ng of input RNA was used to produce cDNA for downstream library preparation. Samples were sequenced on a NextSeq 500 (Illumina) system. Raw reads were aligned, normalized, and further analyzed using Partek Genomics Suite software, version 7.0 Copyright; Partek Inc. Normalized read counts were tested for differential expression using Partek's gene-specific analysis (GSA) algorithm leveraging lognormal with shrinkage[45]. To explore expression gradient and narrow $z$-score range, we separately analyzed CD200R$^-$ and CD200R$^-$ ILC2s. During the analysis of CD200R$^-$ ILC2s, we assigned $z$-scores of zero (black) to all ILC2s without any CD200R mRNA. In a second parallel analysis of CD200R$^+$ ILC2s, we assigned $z$-scores of zero (black) to ILC2s with the lowest non-zero CD200R expression level ($z$-score of 5.22 in our data set) using Partek Flow.

**Supernatant cytokine measurement**. Following manufacturer's instructions, mouse IL-5 and IL-13 ELISA (ThermoFisher Scientific) were utilized to assess cytokines secretion levels. Moreover, 32-plex (MCYTMAG70KPX32) and 41-plex (HCYTMAG-60K-PX41) Millipore Human Cytokine panel kits were used for assessment of additional cytokines, following manufacture specific protocols for multiplexed fluorescent bead-based immunoassay detection (MILLIPLEX® MAP system, Millipore Corporation, Missouri, USA). The curve was completed using different concentrations of the cytokine standards each time, and then evaluated using MasterPlex2012 software (Hitachi Solutions America, Ltd.), as previously described before[32,46].

**Histological analysis of the lungs**. After euthanizing the mice, lungs were harvested and fixed immediately with 4% paraformaldehyde in PBS. After overnight fixation, the lungs were processed for histology. The lung tissue was embedded in paraffin, cut into 4 μm sections and stained with H&E and AB-PAS according to standard protocols. Sections were scanned using light microscope for inflammation. Images of hematoxylin and eosin–stained and alcian blue/periodic acid–schiff–stained tissue slides were acquired with a KeyenceBZ-9000 microscope (Keyence, Itasca, Ill) and assembled into multipaneled figures using Adobe Illustrator software (version 22.1). Histologic images were analyzed with the ImageJ Analysis Application (NIH & LOCI, University of Wisconsin).

**Statistical analysis**. All data are expressed as mean ± standard error of the mean (SEM). A two-tailed Student's $t$-test for unpaired data was applied for comparisons between two groups. For multigroup comparisons, we used one-way ANOVA with the Tukey posthoc test. Data were analyzed with Prism Software (GraphPad Software Inc.). Error bars represent standard error of the mean. $p$ value < 0.05 was considered to denote statistical significance.

**Reporting summary**. Further information on research design is available in the Nature Research Reporting Summary linked to this article.

## Data availability

Sequence data that support the findings of this study have been deposited in Genbank under the primary accession code [GSE166889]. All other data are available in the article and its supplementary files or from the corresponding author upon reasonable request. Source data are provided with this paper.

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

## Acknowledgements

This article was financially supported by National Institutes of Health Public Health Service grants R01 ES025786, R01 ES021801, R01 HL144790, R01 HL133169, R01 HL148110, R01 HL151493, and R01 AI145813 (O.A.). We are grateful to USC Bioinformatics Service for assisting with data analysis. The bioinformatics software and computing resources used in the analysis are funded by the USC Office of Research and the Norris Medical Library. We would like to thank Servier Medical Art for publicly providing the art images of mice, humans, and lungs under a Creative Commons Attribution 3.0 Unported License (https://smart.servier.com/; https://creativecommons.org/licenses/by/3.0/legalcode). These art images were not further changed or modified in this study.

## Author contributions

P.S.J. performed and analyzed all experiments and wrote the manuscript, D.G.H., B.P.H., L.G.T., E.H., C.Q., J.D.P., M.L., and Y.H.E.L helped perform experiments and provided animal husbandry for experiments. O.A. supervised, designed the experiments, conceptualized, interpreted the data and finalized the manuscript. All authors helped with reviewing and editing the manuscript.

## Competing interests

The authors declare no competing interests.
