## [Peer Review File · Nature Communications]

REVIEWER COMMENTS

Reviewer #1 (Remarks to the Author):

This manuscript details the expression of CD200R on ILC2 cells and the effect of its modulation in several models of allergic airways disease. Effects are seen when CD200R is ligated before or after allergic induction. The authors also show relevance to human ILC2s, including in vivo following transfer into mice lacking most immune cells.

CD200R signalling has been associated with the inhibition of inflammation for over 10 years in models ranging from tumour induction, allergy, infection and transplantation. The fact that it reduces pulmonary allergic inflammation is therefore not surprising. The novelty of the current data is the effect of CD200r ligation on ILCs (though a role on ILC1s has been known for some time).

The manuscript takes the reader through a careful and thorough sequence of models to prove their point. In some cases the manuscript could be significantly shortened. Figures 3, 4 and 5, for example could be merged to show only the pertinent results in each model system as the text for these figures is essentially identical.

Though I admire the thoroughness of the approach, some critical areas appear over emphasised. In figure 1 the conclusion is that IL-33 increases CD200R expression on ILC2s. However, with or without IL-33 100% of ILC2s express CD200R - the authors place a lot of emphasis on a small shift in mean fluorescent intensity. This shift is so small that it is unlikely to have biological meaning. This is important, as in the rest of the manuscript, the authors fail to perform a non-IL-33 treated control. Only this control could tell you whether the shift is meaningful.

In addition, it is unclear whether the effects are IL-33 specific since they don't test any other stimuli, or attempt to block the IL-33 receptor.

Line 127 contains a mistake as it refers to augmented cytokine production by CD200R when they mean diminished.

Personally, I find it difficult to interpret the data in figure 1C - a method used throughout the paper. Figure 2 also contains some plots with very small reductions in mean fluorescent intensities of various proteins, a lot of these are very mild effects and so the interpretation should be toned down.

For figure 6h,k,l they state that CD200R ligation reduced pIKK, P65 and p52. Based on the flow plots this is marginal and again over-emphasised. They don't provide raw data for relA/B, but based on the bar charts, it seems that these effects are even more marginal.

Figure 7 examine CD200R on human ILC2s where again they state that CD200R is "highly inducible". This is over-emphasised as all ILC2s express CD200R and IL-33 only marginally increases the intensity per cell.

Overall, a well executed study, but over-interpreted.

Reviewer #2 (Remarks to the Author):

In the manuscript "CD200-CD200R immune checkpoint regulates ILC2 effector function and ameliorates lung inflammation in asthma", Shafiei-Jahani and colleagues investigated the effect of CD200R agonism on lung ILC2 responses and allergic airway disease pathogenesis. The authors determined that CD200R ligation inhibits ILC2 activation and innate allergic airway inflammation in mice. Importantly, the authors used a humanized ILC2 mouse model to confirm findings suggesting the mechanism translates to humans. This report adds to the short list of known ILC2

inhibitors and identifies a potential therapeutic target and compound for the treatment of allergic airway disease. Overall, the work is novel, thorough, shows an impressive phenotype, and is well written. However, I have a few comments that would improve the work if addressed.

Major comments

- 1) Potential plasticity induced via CD200. Given the recent reports of ILC plasticity and regulatory ILCs (ILC210s and ILCregs), the CD200-induced IL-10 production and GATA3 downregulation is very interesting. Do the CD200-stimulated ILC2s express FoxP3 similar ILCregs or lack expression like ILC210s? Is there a shift toward ILC1 and ILC3 phenotypes after CD200 ligation? IL-17 can be produced by "inflammatory" ILC2s as well as ILC3s. Is IL-17 also reduced by CD200 ligation? Do human ILC2s produce IL-10 with CD200R agonism as well?
- 2) Ligand for CD200 in the lung. Identification of the dominant source(s) of CD200L in the challenged lung models would be useful to understand how CD200-expressing ILC2s are regulated in vivo. Is the ligand induced in specific cell types upon IL-33/*Alternaria* stimulation compared with naïve mouse lung?

Minor comments

- 1) Providing PAS lung section staining for mucus production would be helpful to round out the phenotypes.
- 2) The 3-dimensional plots (if not interactive) add little value over 2-dimensional plots and are may be more difficult to interpret.
- 3) The authors mention HDM in their methods but there do not appear to be experiments with HDM. Suggest revising.
- 4) Page 4, line 93. "validate" is misspelled.
- 5) Page 5, line 115. "aILC2s" are not defined as activated ILC2s prior to abbreviation used.

Reviewer #3 (Remarks to the Author):

The present study investigates the CD200-CD200R immunomodulatory axis on type 2 innate lymphoid cells (ILC2s) in the context of allergic asthma. ILC2s are enriched at mucosal sites such as the lung epithelium and constitute early producers of type 2 cytokines during lung inflammation and allergic asthma. Consequently, it is of great interest to decipher the regulatory cues that underly ILC2 activation and cytokine production. Therefore, the CD200R regulatory axis on ILC2s is potentially of great interest, however, the relative contributions and therefore the actual impact of this axis on ILC2s studied within this study remain questionable. Previous studies showed the inhibitory capacity of CD200R ligation on myeloid cells and a lack of CD200R signaling was shown to enhance immune responses. Moreover, CD200R^{-/-} mice showed enhanced bacterial burden in neutrophils during *Francisella tularensis* infection, and mice lacking CD200 had higher macrophage activity during influenza infection. Importantly, the administration of agonists that bind CD200R, however, prevented inflammatory lung disease.

Here, the authors nicely demonstrate the role of CD200R on ILC2 as a checkpoint inhibitor for activation, proliferation and type 2 cytokine secretion. The authors also translate their findings to human derived ILC2 making a translational approach. In general, the manuscript is well written, the experiments are well performed and data are well presented. However, the data does not support the strong message of the title and the manuscript as the authors did not experimentally rule out the involvement of other CD200R expression cells (macrophages, neutrophils, mast cells etc...) potentially responsible for the observed phenotypes.

Other comments:

- 1) Results described in the text are difficult to see in the graphs shown in Fig. 1c. It is stated that aILC2s with high CD200R expression have reduced total gene expression, as indicated by color and dot size, respectively. However, it is difficult for the reader to appreciate this correlation. A correlation graph showing CD200R expression levels versus total genes expressed could resolve this issue.
- 2) There is a striking discrepancy regarding CD200R expression in that mRNA expression seems to

be highly heterogeneous within aILC2s (Fig. 1c), whereas it was shown to be homogeneously expressed at protein levels (Fig. 1d).

3) The scaling of Cd200r1 expression seems biased, as it does not visibly distinguish expression levels, despite covering a seemingly high range of expression or z-scores (Fig 1c). In other words: everything unequal to zero is depicted as green.

4) The inhibitory role of CD200R signaling on macrophages has previously been shown in context of an influenza model. Since macrophages also play an important role during allergen hyperreactivity, the relative contribution of ILC2s to the observed phenotype remains unanswered.

5) Regarding point 3) and 4) it would be interesting to compare ILC2 and macrophages mRNA and protein levels of CD200R.

Minor points:

1) The manuscript entails a wealth of spelling mistakes.

2) The discussion is almost a review.

3) It seems inconsistent that from Fig2 on the authors did not show neutrophil numbers in BAL anymore.

4) ScRNA-seq data is often shown as 3D graphs with specific expression indicated as a color and total genes expressed indicated as size of the dots. In addition, these plots are marked with dashed lines. However, it is not mentioned what these lines are indicating.

5) NFkB signaling was assessed by flow cytometry (Fig 6h-i). To validate these results controls (positive and negative) should be added to the experiments.

6) The authors do not show IL-10 expression by human aILC2s with and without CD200 FC (Fig. 7d). Since they emphasize the strong translational potential and IL-10 secretion was enhanced in the murine setting, IL-10 production should also be assessed with human aILC2s.

We thank the editors and reviewers for their constructive feedback on our manuscript. In response to these comments, we have made changes to the manuscript, including performing additional experiments and analyses that have been included in the revised figures and supplementary figures. We believe that these changes have helped clarify and improve our work. Please see our point-by-point response below.

Reviewer #1 (Remarks to the Author):

This manuscript details the expression of CD200R on ILC2 cells and the effect of its modulation in several models of allergic airways disease. Effects are seen when CD200R is ligated before or after allergic induction. The authors also show relevance to human ILC2s, including in vivo following transfer into mice lacking most immune cells. CD200R signaling has been associated with the inhibition of inflammation for over 10 years in models ranging from tumour induction, allergy, infection and transplantation. The fact that it reduces pulmonary allergic inflammation is therefore not surprising. The novelty of the current data is the effect of CD200r ligation on ILCs (though a role on ILC1s has been known for some time).

The manuscript takes the reader through a careful and thorough sequence of models to prove their point. In some cases the manuscript could be significantly shortened. Figures 3, 4 and 5, for example could be merged to show only the pertinent results in each model system as the text for these figures is essentially identical.

We understand the reviewer's concern regarding the length of our manuscript. We have now significantly shortened the results and moved the results of experiments with *Alternaria alternata* to **Supplementary Figure 6**. Additionally, we now condensed the text in the result section.

Though I admire the thoroughness of the approach, some critical areas appear over emphasized. In figure 1 the conclusion is that IL-33 increases CD200R expression on ILC2s. However, with or without IL-33 100% of ILC2s express CD200R - the authors place a lot of emphasis on a small shift in mean fluorescent intensity. This shift is so small that it is unlikely to have biological meaning. This is important, as in the rest of the manuscript, the authors fail to perform a non-IL-33 treated control. Only this control could tell you whether the shift is meaningful.

We agree with the reviewer's comment that CD200R is expressed at a high basal level on ILC2s. In response to reviewer 3 below, we now additionally assess the basal expression level of CD200R among ILC2s, and compare the results to expression levels on pulmonary macrophages. Our results suggested that at steady state both ILC2s and macrophages express comparable levels of CD200R; however, after activation by IL-33 only ILC2s exhibited significantly higher CD200R expression. The results of these experiments are now added as **Supplementary Figure 1**.

We need to point out that the CD200R expression, assessed by flow cytometry, is displayed on a logarithmic scale (**Figure 1d, 1f, 6d, and Supplementary Figures 2a**). Therefore, although it may appear as a small shift, the expression change is statistically and biologically significant. We believe the upregulation of CD200R on activated ILC2s during lung inflammation may significantly contribute to amelioration of AHR that we observed after treatment with CD200 FC. We carefully discuss this further in the manuscript. Unfortunately, it is not technically feasible to study the role of inhibitory receptors, such as CD200R on naïve ILC2s as the level of cytokines at steady is relatively low. We now discuss the limitation of our studies in the discussion section of the manuscript.

In addition, it is unclear whether the effects are IL-33 specific since they don't test any other stimuli, or attempt to block the IL-33 receptor.

This is an interesting question. We performed additional experiments utilizing an anti-ST2 (IL-33 receptor) blocking antibody. Briefly, we assessed the expression levels of CD200R in presence of anti-ST2 blocking antibody or isotype control on ILC2s stimulated with IL-33 or PBS. Our results demonstrate that blocking IL-33/ST2 axis prevented the induction of CD200R on ILC2s. These results are now added as **Supplementary Figure 2**.

Line 127 contains a mistake as it refers to augmented cytokine production by CD200R when they mean diminished.

We have corrected the text accordingly.

Personally, I find it difficult to interpret the data in figure 1C - a method used throughout the paper. Figure 2 also contains some plots with very small reductions in mean fluorescent intensities of various proteins, a lot of these are very mild effects and so the interpretation should be toned down.

We understand the reviewer's concern around clarity of scRNA sequencing data and have now carefully revised the text to precisely reflect the results presented. Additionally, we consulted USC biostatisticians and now analyze these data as regression curve, which is much easier to interpret (**Supplementary Figure 3a-b**). Consistent with our previous findings (**Figure 1c**), these results clearly demonstrate CD200R expression is inversely correlated with total gene expression. Please also note our bulk RNA sequencing results in **Figure 5b** also confirms these results and show that CD200R engagement reduces total gene expression in activated ILC2s.

For figure 6h,k,l they state that CD200R ligation reduced pIKK, P65 and p52. Based on the flow plots this is marginal and again over-emphasised. They don't provide raw data for RelA/B, but based on the bar charts, it seems that these effects are even more marginal.

In order to decipher the molecular mechanisms associated CD200R-dependent inhibition of ILC2s, we previously quantified transcriptomic landscape of ILC2s treated with CD200 FC or isotype control. Our RNA sequencing data revealed that CD200R engagement on ILC2s modulated the gene expression of the transcriptional factors that are involved in both canonical and non-canonical NF- κ B pathways. In order to further validate these mRNA results, we next measured the components of these two pathways at protein level utilizing flow cytometry. It is important to note we measured the phosphorylated forms (pIKK α , p52 and p65) and displayed the results on a logarithmic scale. Prior studies have clearly demonstrated that these shifts, although they may appear small, have significant biological impact and small amounts of phosphorylation can initiate a cascade of events that severely alters the transcriptional landscape and effector function of immune cells¹⁻⁵. We do apologize that we were not clear about the experiments regarding RelA/B presented as **Figure 5i** and **5j**. RelA and RelB show significant upregulation in our RNAseq pathway analysis, therefore, we felt it is essential to provide the raw data for the actual mRNA counts. Additionally, following the journal's guidelines

all raw RNA sequencing results will be made publicly available via the Gene Expression Omnibus (GEO) database online.

Figure 7 examine CD200R on human ILC2s where again they state that CD200R is "highly inducible". This is over-emphasised as all ILC2s express CD200R and IL-33 only marginally increases the intensity per cell. Overall, a well executed study, but over-interpreted.

We thank the reviewer for their constructive feedback. We now tone down the interpretation of our results in the manuscript.

Reviewer #2 (Remarks to the Author):

In the manuscript "CD200–CD200R immune checkpoint regulates ILC2 effector function and ameliorates lung inflammation in asthma", Shafiei-Jahani and colleagues investigated the effect of CD200R agonism on lung ILC2 responses and allergic airway disease pathogenesis. The authors determined that CD200R ligation inhibits ILC2 activation and innate allergic airway inflammation in mice. Importantly, the authors used a humanized ILC2 mouse model to confirm findings suggesting the mechanism translates to humans. This report adds to the short list of known ILC2 inhibitors and identifies a potential therapeutic target and compound for the treatment of allergic airway disease. Overall, the work is novel, thorough, shows an impressive phenotype, and is well written. However, I have a few comments that would improve the work if addressed.

Major comments

1) Potential plasticity induced via CD200. Given the recent reports of ILC plasticity and regulatory ILCs (ILC210s and ILCregs), the CD200-induced IL-10 production and GATA3 downregulation is very interesting. Do the CD200-stimulated ILC2s express FoxP3 similar ILCregs or lack expression like ILC210s? Is there a shift toward ILC1 and ILC3 phenotypes after CD200 ligation? IL-17 can be produced by "inflammatory" ILC2s as well as ILC3s. Is IL-17 also reduced by CD200 ligation? Do human ILC2s produce IL-10 with CD200R agonism as well?

The reviewer has raised an interesting point regarding potential plasticity of ILC2s after CD200R engagement. We performed additional experiments and assessed the expression levels of transcription factors specific for ILC1s and ILC3s⁶. Our analysis by flow cytometry revealed that CD200R signaling does not shift ILC2s toward either an ILC1 or ILC3 phenotype, as evident by lack of T-bet and ROR γ t expression. The results of these experiments are now added as **Supplementary Figure 4a** and **4b**.

Our group shares the reviewer's interest in better understanding the different regulatory phenotypes reported in subsets of ILC2s. We recently reported that expression of IL-10 by ILC2₁₀s is dependent on transcription factors Blimp-1, cMaf, and ILC2₁₀s do not express FOXP3⁷. Therefore, we carefully assessed the expression levels of above-mentioned transcription factors in IL-33-activated ILC2s treated with CD200 FC or isotype control. Our results suggest that CD200R engagement significantly upregulates expression of Blimp-1. However, CD200R signaling failed to induce cMaf or FOXP3 expression. These results suggest IL-10 production via CD200R engagement leverages the Blimp-1 transcriptional factor. We now added the results of these experiments as **Supplementary Figure 4c-e** and discuss their implications in the manuscript.

In murine models, IL-10 is reportedly upregulated by the engagement of gamma c receptor (IL-2 and IL-4 signaling pathways)^{7,8}. Here we reported CD200R engagement on murine ILC2s can induce Blimp-1. However, IL-10 production in human ILC2s has remained a challenging area of research. Although IL-10 producing ILC2s have been observed in humans samples, their specific biology and induction mechanism has largely remained a mystery^{9,10}. We have previously attempted to induce an IL-10⁺ phenotype in human ILC2s via engagement of gamma c receptor (via IL-2 and IL-4); as well as CD200R signaling pathways. However, these attempts have unfortunately not been successful, suggesting induction of IL-10 in human ILC2s may require different signaling pathways that warrant additional studies. Moreover, we need to point out murine ILC2s were isolated from lung but the human ILC2s were isolated from PBMCs; therefore, the tissue specific signature of ILC2 may also contribute to production of cytokines such as IL-10. Finally, it is important to point out that our results demonstrate CD200R engagement on human ILC2s reduces AHR in the humanized murine models, suggesting that CD200R signaling reduces type 2 cytokines production and alleviates AHR independent of IL-10. We now discuss this further in the manuscript.

2) Ligand for CD200 in the lung. Identification of the dominant source(s) of CD200L in the challenged lung models would be useful to understand how CD200-expressing ILC2s are regulated in vivo. Is the ligand induced in specific cell types upon IL-33/Alternaria stimulation

Regulation of ILC2s particularly in biological system is an important topic of interest. In order to evaluate the cellular landscape and potential sources of the CD200 ligand, we have analyzed single cell transcriptome obtained from lung samples of healthy donors and asthmatics, as described before¹¹. Briefly, these results demonstrate that there are multiple sources of CD200 within the human lungs, including smooth muscle cells, endothelial cells, submucosal cells, ciliated and mucous ciliated cells. Interestingly, CD200 is upregulated in at least 8 pulmonary cells types, including fibroblasts and T cells in patients with asthma. These results are now added as **Figure 6a** and discussed further in the manuscript.

Minor comments

1) Providing PAS lung section staining for mucus production would be helpful to round out the phenotypes.

We have performed additional experiments and stained the lungs with alcian blue/periodic acid–schiff (AB-PAS) staining (**Figure 3**, **Figure 4**, and **Supplementary Figure 6**). These new results support the notion that anti-CD200R treatment significantly reduces goblet cell hyperplasia and mucus secretion.

2) The 3-dimensional plots (if not interactive) add little value over 2-dimensional plots and are may be more difficult to interpret.

We understand the review's concern, and have now performed additional analysis (**Supplementary Figure 3**) to better present the interpretations of the tSNE plots. Briefly, these results clearly demonstrate CD200R expression is inversely correlated with total gene expression and provide an alternative plot to portray CD200R expression gradient on ILC2s.

3) The authors mention HDM in their methods but there do not appear to be experiments with HDM. Suggest revising.

We apologize and have now corrected this error in the manuscript.

4) Page 4, line 93. "validate" is misspelled.

We have corrected the typo within text.

5) Page 5, line 115. "aILC2s" are not defined as activated ILC2s prior to abbreviation used.

We now define aILC2s as IL-33 activated ILC2s in the manuscript.

Reviewer #3 (Remarks to the Author):

The present study investigates the CD200-CD200R immunomodulatory axis on type 2 innate lymphoid cells (ILC2s) in the context of allergic asthma. ILC2s are enriched at mucosal sites such as the lung epithelium and constitute early producers of type 2 cytokines during lung inflammation and allergic asthma. Consequently, it is of great interest to decipher the regulatory cues that underly ILC2 activation and cytokine production. Therefore, the CD200R regulatory axis on ILC2s is potentially of great interest, however, the relative contributions and therefore the actual impact of this axis on ILC2s studied within this study remain questionable. Previous studies showed the inhibitory capacity of CD200R ligation on myeloid cells and a lack of CD200R signaling was shown to enhance immune responses. Moreover, CD200R^{-/-} mice showed enhanced bacterial burden in neutrophils during *Francisella tularensis* infection, and mice lacking CD200 had higher macrophage activity during influenza infection. Importantly, the administration of agonists that bind CD200R, however, prevented inflammatory lung disease. Here, the authors nicely demonstrate the role of CD200R on ILC2 as a checkpoint inhibitor for activation, proliferation and type 2 cytokine secretion. The authors also translate their findings to human derived ILC2 making a translational approach. In general, the manuscript is well written, the experiments are well performed and data are well presented. However, the data does not support the strong message of the title and the manuscript as the authors did not experimentally rule out the involvement of other CD200R expression cells (macrophages, neutrophils, mast cells etc...) potentially responsible for the observed phenotypes.

Other comments:

1) Results described in the text are difficult to see in the graphs shown in Fig. 1c. It is stated that aILC2s with high CD200R expression have reduced total gene expression, as indicated by color and dot size, respectively. However, it is difficult for the reader to appreciate this correlation. A correlation graph showing CD200R expression levels versus total genes expressed could resolve this issue.

We appreciate the review's suggestion and have now included correlation graphs depicting CD200R expression levels versus total genes expressed on naïve (**Supplementary Figure 3a**) and activated ILC2s (**Supplementary Figure 3b**). Consistent with our previous results, these new plots clearly demonstrate that higher CD200R expression is correlated with reduced total gene expression in pulmonary ILC2s. We discuss these new results further in the manuscript.

2) There is a striking discrepancy regarding CD200R expression in that mRNA expression seems to be highly heterogeneous within aILC2s (Fig. 1c), whereas it was shown to be homogeneously expressed at protein levels (Fig. 1d)

We agree with the reviewer that CD200R expression pattern at a mRNA level appears to be different than the protein level. Since mRNA is ultimately translated into protein, mRNA and protein levels are typically presumed correlated. However, dichotomies between mRNA and proteins expression patterns have been previously observed for many genes, which can manifest from a myriad of factors that are not mutually exclusive¹². For instance, the post-transcriptional mechanisms involved in turning mRNA into protein that are not yet sufficiently understood to be able to predict protein concentrations from mRNA^{12,13}. Similarly, the transcription and translation kinetics may diverge considerably, as mRNA typically has reduced stability and longevity compared to the protein produced¹³⁻¹⁶. We now discuss this further in the manuscript.

3) The scaling of Cd200r1 expression seems biased, as it does not visibly distinguish expression levels, despite covering a seemingly high range of expression or z-scores (Fig 1c). In other words: everything unequal to zero is depicted as green

We apologize for the lack of clarity in presenting our results. While leveraging single cell RNA sequencing, we observed a large CD200R expression landscape across all ILC2s. We initially assigned relative CD200R expression values, and normalized all the scores to the lowest existing values (ILC2s without any CD200R mRNA expression were assigned z-scores of zero and appeared as black). Nevertheless, we agree with the reviewer's point that the large z-score landscape leads to difficulty appreciating the color gradient and have performed additional analysis to better show the CD200R mRNA expression gradient. Briefly, we separately analyzed CD200R⁺ and CD200R⁻ ILC2s. During the analysis of CD200R⁻ ILC2s, we assigned z-scores of zero (black) to all ILC2s without any CD200R mRNA. In a second parallel analysis of CD200R⁺ ILC2s, we assigned z-scores of zero (black) to ILC2s with the lowest (non-zero) CD200R expression level (z-score of 5.22 in our data set). Since the highest relative expression value was still ranked with a z-score of 11.26, the separation of analysis narrowed the range within the expression landscape (5.22 to 11.26) among CD200R⁺ ILC2s. As evident by the color gradient in **Supplementary Figure 3c**, CD200R expression follows a gradual increase and visibly distinguish levels among CD200R⁺ cells. Interestingly, the gap between 0 to 5.22 suggest that once the cell is instructed to express the gene *CD200r1*, there is a high basal level of mRNA production. We now clarify discuss the rationale behind of analytic methods; as well as the biological implications of these new findings within the manuscript.

4) The inhibitory role of CD200R signaling on macrophages has previously been shown in context of an influenza model. Since macrophages also play an important role during allergen hyperreactivity, the relative contribution of ILC2s to the observed phenotype remains unanswered.

We agree with the reviewer that anti-CD200R treatments will also target other CD200R⁺ cells, such as macrophages. However, please note in our humanized mice experiments we utilized human CD200 FC that only binds and engages human CD200R. Therefore, our humanized mice results (**Figure 7**) demonstrate *in vivo* CD200R engagement on human ILC2s in alymphoid recipients is sufficient for amelioration of ILC2-driven AHR and significant reduction of eosinophilia. We now clarify this information in the manuscript.

5) Regarding point 3) and 4) it would be interesting to compare ILC2 and macrophages mRNA and protein levels of CD200R.

We have performed additional experiments to compare the CD200R expression on ILC2s and macrophages in the lungs. Briefly, a cohort of WT mice were challenged with the IL-33 or PBS intranasally (i.n.) for three consecutive days. On the fourth day, macrophages and ILC2s from the lungs were isolated and analyzed by flow cytometry. These results demonstrate that ILC2s and macrophages display a comparable CD200R expression at steady state (**Supplementary Figure 1**). Interestingly, CD200R expression is inducible by IL-33 in ILC2s, and is not induced in macrophages. Additionally, we now use a blocking antibody and demonstrate that CD200R induction on ILC2s is dependent on engagement of ST2 by IL-33 (**Supplementary Figure 2**), as described in response to the reviewer 1 above. We now discuss these new results, and compare CD200R expression on ILC2s and macrophages in the manuscript.

Minor points:

1) The manuscript entails a wealth of spelling mistakes.

We have now carefully proofread the manuscript.

2) The discussion is almost a review.

We have modified the discussion and incorporated our findings in relation to prior studies.

3) It seems inconsistent that from Fig2 on the authors did not show neutrophil numbers in BAL anymore.

As requested, we have included the number of BAL neutrophils in our preventative and therapeutic IL-33 (**Figure 3** and **Figure 4**) and *Alternaria alternata* (**Supplementary Figure 6**) models.

4) ScRNA-seq data is often shown as 3D graphs with specific expression indicated as a color and total genes expressed indicated as size of the dots. In addition, these plots are marked with dashed lines. However, it is not mentioned what these lines are indicating.

The enclosed cells by the dashed line have the highest CD200R expression and the least total gene expression. We now clarify this information within the text.

5) NFκB signaling was assessed by flow cytometry (Fig 6h-i). To validate these results controls (positive and negative) should be added to the experiments.

In order to decipher the molecular mechanisms associated CD200R-dependent inhibition of ILC2s, our RNA sequencing data revealed modulation of the transcriptional factors involved in both canonical and non-canonical NF-κB pathways. In order to further validate these mRNA results, we next measured the components of these two pathways at protein level using flow cytometry. Specifically, we measured the phosphorylated factors (pIKKα, p52 and p65) and leveraged the appropriate controls including isotype and staining controls, as demonstrated by our group before¹⁷⁻¹⁹. We now clarify our protocol and the controls used within the manuscript.

6) The authors do not show IL-10 expression by human aILC2s with and without CD200 FC (Fig. 7d). Since they emphasize the strong translational potential and IL-10 secretion was enhanced in the murine setting, IL-10 production should also be assessed with human aILC2s.

In murine ILC2s, our group and others have previously reported IL-10 is upregulated by the engagement of gamma c receptor (IL-2 and IL-4 signaling pathways)^{7,8}. Here we also reported CD200R engagement on murine ILC2s can induce IL-10. IL-10 producing ILC2s have been observed in humans samples; however their specific biology and induction mechanism remain to be elucidated^{9,10}. We have previously attempted to induce an IL-10⁺ phenotype in human ILC2s via engagement of gamma c receptor (via IL-2 and IL-4); as well as CD200R signaling pathways. However, these attempts have unfortunately not been successful, suggesting induction of IL-10 in human ILC2s may require different signaling pathways. Moreover, murine ILC2s in our study were isolated from the lung, whereas the human ILC2s were isolated from PBMCs. As a result, the tissue specific signature of ILC2 may also contribute to production of cytokines such as IL-10. We now discuss this limitation further in the manuscript.

Importantly, our humanized models demonstrate that CD200R engagement on human ILC2s reduces pulmonary inflammation, suggesting that CD200R signaling can reduce type 2 cytokines production and alleviate AHR independent of IL-10 production.

References

- 1 Christian, F., Smith, E. L. & Carmody, R. J. The Regulation of NF-kappaB Subunits by Phosphorylation. *Cells* **5**, doi:10.3390/cells5010012 (2016).
- 2 Adli, M., Merkhofer, E., Cogswell, P. & Baldwin, A. S. IKKalpha and IKKbeta each function to regulate NF-kappaB activation in the TNF-induced/canonical pathway. *PLoS One* **5**, e9428, doi:10.1371/journal.pone.0009428 (2010).
- 3 Yamamoto, Y., Yin, M. J. & Gaynor, R. B. IkappaB kinase alpha (IKKalpha) regulation of IKKbeta kinase activity. *Mol Cell Biol* **20**, 3655-3666, doi:10.1128/mcb.20.10.3655-3666.2000 (2000).
- 4 Oeckinghaus, A. & Ghosh, S. The NF-kappaB family of transcription factors and its regulation. *Cold Spring Harb Perspect Biol* **1**, a000034, doi:10.1101/cshperspect.a000034 (2009).
- 5 Kabata, H., Moro, K. & Koyasu, S. The group 2 innate lymphoid cell (ILC2) regulatory network and its underlying mechanisms. *Immunol Rev* **286**, 37-52, doi:10.1111/imr.12706 (2018).
- 6 Colonna, M. Innate Lymphoid Cells: Diversity, Plasticity, and Unique Functions in Immunity. *Immunity* **48**, 1104-1117, doi:10.1016/j.immuni.2018.05.013 (2018).
- 7 Howard, E. *et al.* IL-10 production by ILC2s requires Blimp-1 and cMaf, modulates cellular metabolism, and ameliorates airway hyperreactivity. *J Allergy Clin Immunol*, doi:10.1016/j.jaci.2020.08.024 (2020).
- 8 Bando, J. K. *et al.* ILC2s are the predominant source of intestinal ILC-derived IL-10. *J Exp Med* **217**, doi:10.1084/jem.20191520 (2020).
- 9 Bonne-Annee, S., Bush, M. C. & Nutman, T. B. Differential Modulation of Human Innate Lymphoid Cell (ILC) Subsets by IL-10 and TGF-beta. *Sci Rep* **9**, 14305, doi:10.1038/s41598-019-50308-8 (2019).
- 10 Wang, S. *et al.* Regulatory Innate Lymphoid Cells Control Innate Intestinal Inflammation. *Cell* **171**, 201-216 e218, doi:10.1016/j.cell.2017.07.027 (2017).
- 11 Vieira Braga, F. A. *et al.* A cellular census of human lungs identifies novel cell states in health and in asthma. *Nat Med* **25**, 1153-1163, doi:10.1038/s41591-019-0468-5 (2019).

- 12 Liu, Y., Beyer, A. & Aebersold, R. On the Dependency of Cellular Protein Levels on mRNA Abundance. *Cell* **165**, 535-550, doi:10.1016/j.cell.2016.03.014 (2016).
- 13 Boo, S. H. & Kim, Y. K. The emerging role of RNA modifications in the regulation of mRNA stability. *Exp Mol Med* **52**, 400-408, doi:10.1038/s12276-020-0407-z (2020).
- 14 Shao, W. *et al.* Comparative analysis of mRNA and protein degradation in prostate tissues indicates high stability of proteins. *Nat Commun* **10**, 2524, doi:10.1038/s41467-019-10513-5 (2019).
- 15 Stein, K. C. & Frydman, J. The stop-and-go traffic regulating protein biogenesis: How translation kinetics controls proteostasis. *J Biol Chem* **294**, 2076-2084, doi:10.1074/jbc.REV118.002814 (2019).
- 16 Chan, L. Y., Mugler, C. F., Heinrich, S., Vallotton, P. & Weis, K. Non-invasive measurement of mRNA decay reveals translation initiation as the major determinant of mRNA stability. *Elife* **7**, doi:10.7554/eLife.32536 (2018).
- 17 Shafiei-Jahani, P. *et al.* DR3 stimulation of adipose resident ILC2s ameliorates type 2 diabetes mellitus. *Nat Commun* **11**, 4718, doi:10.1038/s41467-020-18601-7 (2020).
- 18 Galle-Treger, L. *et al.* Costimulation of type-2 innate lymphoid cells by GITR promotes effector function and ameliorates type 2 diabetes. *Nat Commun* **10**, 713, doi:10.1038/s41467-019-08449-x (2019).
- 19 Hurrell, B. P. *et al.* TNFR2 Signaling Enhances ILC2 Survival, Function, and Induction of Airway Hyperreactivity. *Cell Rep* **29**, 4509-4524 e4505, doi:10.1016/j.celrep.2019.11.102 (2019).

REVIEWERS' COMMENTS

Reviewer #1 (Remarks to the Author):

I have re-reviewed the manuscript that examines the effect of CD200R ligation on ILC2s. I have also examined the comments from the other reviewers. The authors have addressed most of the concerns raised and have improved the manuscript

Reviewer #2 (Remarks to the Author):

The authors have adequately addressed my comments.

Reviewer #3 (Remarks to the Author):

I thank the authors for the clarification.

However, since Rag^{-/-}Il2r^{-/-} do not lack other CD220R⁺ cells like macrophages, it is necessary to show that CD220 FC does not cross react with murine CD200R or at least provide a reference to such data.

Beside that, the authors have adequately addressed the rest of our comments and we would recommend to accept this manuscript if this remaining issue is addressed.

Reviewer #1 (Remarks to the Author):

I have re-reviewed the manuscript that examines the effect of CD200R ligation on ILC2s. I have also examined the comments from the other reviewers. The authors have addressed most of the concerns raised and have improved the manuscript

We thank the reviewer for their positive feedback.

Reviewer #2 (Remarks to the Author):

The authors have adequately addressed my comments.

We appreciate the reviewer's positive feedback.

Reviewer #3 (Remarks to the Author):

I thank the authors for the clarification.

However, since Rag-/-Il2r-/- do not lack other CD220R+ cells like macrophages, it is necessary to show that CD220 FC does not cross react with murine CD200R or at least provide a reference to such data.

We now provide the appropriate references that show human CD200 FC does not cross react with murine CD200R on page 16 within the manuscript¹⁻³.

Beside that, the authors have adequately addressed the rest of our comments and we would recommend to accept this manuscript if this remaining issue is addressed.

We thank the reviewer for their positive feedback.

References

- 1 Wright, G. J. *et al.* Characterization of the CD200 receptor family in mice and humans and their interactions with CD200. *J Immunol* **171**, 3034-3046, doi:10.4049/jimmunol.171.6.3034 (2003).
- 2 Hatherley, D., Lea, S. M., Johnson, S. & Barclay, A. N. Structures of CD200/CD200 receptor family and implications for topology, regulation, and evolution. *Structure* **21**, 820-832, doi:10.1016/j.str.2013.03.008 (2013).
- 3 Hatherley, D. & Barclay, A. N. The CD200 and CD200 receptor cell surface proteins interact through their N-terminal immunoglobulin-like domains. *Eur J Immunol* **34**, 1688-1694, doi:10.1002/eji.200425080 (2004).